

# The Critical Role of Aqueous-Phase Processes in Aromatic-Derived Nitrogen-Containing Organic Aerosol Formation in Cities with Different Energy Consumption Patterns

Yi-Jia Ma[1,2,3], Yu Xu[2,3]*, Ting Yang[1,2,3], Lin Gui[1,2,3], Hong-Wei Xiao[2,3], Hao Xiao[2,3], and Hua-Yun Xiao[2,3]

[1]School of Environmental Science and Engineering, Shanghai Jiao Tong University, Shanghai 200240, China

[2]School of Agriculture and Biology, Shanghai Jiao Tong University, Shanghai 200240, China

[3]Shanghai Yangtze River Delta Eco-Environmental Change and Management Observation and Research Station, Ministry of Science and Technology, Ministry of Education, Shanghai 200240, China

Yu Xu (E-mail: xuyu360@sjtu.edu.cn)

+8615885507087

Shanghai Jiao Tong University, 800 Dongchuan Road



**Abstract.** Nitrogen-containing organic compounds (NOCs) impact air quality and
human health. Here, the abundance, potential precursors, and main formation
mechanisms of NOCs in PM$_{2.5}$ during winter were compared for the first time among
Haerbin (coal-dependent for heating), Beijing (natural gas and coal as heating energy),
and Hangzhou (no centralized heating policy). The total signal intensity of CHON+,
CHN+, and CHON− compounds was highest in Haerbin and lowest in Hangzhou.
Anthropogenic aromatics accounted for 73%–93% of all identified precursors of
CHON+, CHN+, and CHON− compounds in Haerbin. Although the abundance of
aromatics-derived NOCs was lower in Beijing than in Haerbin, aromatics were also
the main contributors to NOC formation in Beijing. Hangzhou exhibited the lowest
levels of aromatic precursors. Furthermore, non-metric multidimensional scaling
analysis indicated an overall reduction in the impact of fossil fuel combustion on
NOC pollution along the route from Haerbin to Beijing to Hangzhou. We found that
aqueous-phase processes (mainly condensation, hydrolysis or dehydration processes
for reduced NOCs, and mainly oxidization or hydrolysis processes for oxidized NOCs)
can promote the transformation of precursors to produce NOCs, leading to the most
significant increase in aromatic NOC levels in Haerbin (particularly on haze days).
Reduced precursor emissions in Beijing and Hangzhou (the lowest) constrained the
aqueous-phase formation of NOCs. The overall results suggest that the aerosol NOC
pollution in coal-dependent cities is mainly controlled by anthropogenic aromatics





and aqueous-phase processes. Thus, without effective emission controls, the
formation of NOCs through aqueous-phase processes may still pose a large threat to
air quality.
**Keywords:** Aerosols, Nitrogen-containing organic compounds, Heating energy
consumption, Anthropogenic pollutants, Formation mechanism



## 1. Introduction

Nitrogen-containing organic compounds (NOCs) are abundant reactive nitrogen
species in aerosol particles, accounting for up to 40%–80% of total nitrogen
deposition (Li et al., 2023; Xi et al., 2023; Yu et al., 2020). Clearly, aerosol NOCs can
significantly contribute to the global nitrogen cycle (Li et al., 2023; Cape et al., 2011).
Moreover, the formation of secondary organic aerosols (SOA) and light-absorbing
organic aerosols (e.g., brown carbon) is also tightly associated with NOCs (Wang et
al., 2024; Liu et al., 2023b; Zeng et al., 2021), thus affecting the radiative balance and
air quality (Yuan et al., 2023; Jiang et al., 2023). In particular, certain NOCs, such as
nitroaromatics and peroxyacyl nitrates, are characterized as phytotoxins and potential
carcinogens, posing threats to ecosystems and human health (Shi et al., 2023; Singh
and Kumar, 2022; Huang et al., 2024). Therefore, understanding the characteristics,
origins, and atmospheric processes of NOCs is essential for comprehending their
climate and health effects.
Aerosol NOCs can be derived from primary emissions associated with
anthropogenic activities and natural sources (Cape et al., 2011; Lin et al., 2023; Xu et
al., 2020a; Wang et al., 2017; Song et al., 2018; Song et al., 2022; Ma et al., 2024).
Secondary formation processes may play a more crucial role in the formation of
NOCs in fine aerosol particles, which involve interactions among volatile organic
compounds (VOCs), atmospheric oxidants, and reactive inorganic nitrogen species



(Montoya-Aguilera et al., 2018; Perraud et al., 2012; Hallquist et al., 2009). For
instance, laboratory studies have observed the formation of organic nitrates from the
oxidation of isoprene and α-/β-pinene by atmospheric oxidants and nitrogen oxide
($NO_x$) (Surratt et al., 2010; Rollins et al., 2012; Nguyen et al., 2015). Additionally,
aqueous-phase reactions of $NH_4^+$ (or $NH_3$) with biogenic VOCs or carbonyl
compounds have been suggested to be important mechanisms of reduced NOC (Re-
NOCs) formation (Abudumutailifu et al., 2024; Laskin et al., 2014; Li et al., 2019b;
Liu et al., 2023b; Wang et al., 2024). However, understanding the origins, formation
mechanisms, and environmental impacts of NOCs is hindered by the elusive and
intractable molecular information regarding NOCs and their precursors.

Aerosol liquid water (ALW) can greatly increase the formation of aerosol NOCs

by facilitating the conversion of water-soluble organic gases into particles and
subsequently enabling aqueous-phase reactions (Li et al., 2019a; Lv et al., 2022; Liu
et al., 2023b). A positive correlation between aerosol NOC abundance and either ALW
or relative humidity (RH) has been observed in several observation studies (Jiang et
al., 2023; Liu et al., 2023b; Xu et al., 2020b). In particular, it has been suggested that
increased ALW levels can exacerbate winter haze in China (Wu et al., 2018; Hodas et
al., 2014; Lv et al., 2022; Wang et al., 2021d; Liu et al., 2023b; Wang et al., 2021a; Li
et al., 2019a). Presumably, precursors and ALW are the two key factors in the
formation of aerosol NOCs. Haze environments have potentially high RH levels and



large emissions of NOC precursors. Moreover, in Chinese cities with different energy
consumption (e.g., coal, biomass, and natural gas) for winter heating (Zhang et al.,
2021b; Zhang et al., 2023b), the types and emission intensities of pollutants released
from different heating sources are expected to vary considerably (Bond et al., 2006;
Stockwell et al., 2015; Křůmal et al., 2019). However, the potential effects of ALW in
the formation of NOCs in Chinese cities with different energy consumption during
winter, particularly in haze periods, are not well documented. Moreover, the roles of
ALW-related NOC formation processes in the formation of haze in cities with
different energy consumption types also remain largely unknown.

In this study, we present the measurements of the NOCs and other chemical

compositions in PM$_{2.5}$ collected from three cities (Haerbin, Beijing, and Hangzhou)
with different energy consumption during winter. The specific objectives of this study
were: (1) to investigate the differences in the abundance, composition, and major
precursors of NOCs in different cities with different energy consumption, especially
on polluted days; and (2) to elucidate the potential effects of ALW on the formation of
oxidized NOCs (Ox-NOCs) and reduced NOCs (Re-NOCs) during winter
(particularly on polluted days) in cities with different energy consumption. The
research findings are expected to provide valuable implications for the mitigation of
aerosol NOCs pollution in urban environments.





**2. Materials and methods**
**2.1. Study site description and sample collection**
The study sites are located in three urban areas, including Haerbin (HEB, i.e.,
Harbin, 126.64°E, 45.77°N), Beijing (BJ, 116.41°E, 40.04°N), and Hangzhou (HZ,
120.16°E, 30.30°N) (**Fig. S1**). The city of HEB, with a population density of 9.95
million, is situated in the northeastern region of China. It relies heavily on coal for
centralized heating during winter. The rapid urbanization and increased coal
consumption have significantly deteriorated air quality in HEB in recent years (Ma et
al., 2020). In contrast, BJ has largely shifted towards the utilization of cleaner energy
sources (e.g., natural gas) for centralized heating in recent years, particularly
following the implementation of the "Beijing 2013–2017 Clean Air Action Plan" (Vu
et al., 2019; Yuan et al., 2023). HZ, situated within the Yangtze River Delta, is exempt
from the necessity of heating due to the relatively mild winter climate (average
temperature of 6.6 ± 2.4 °C during the sampling period, **Table S1**). Clearly, the
distinctive energy consumption patterns observed in these three cities during winter
provide a valuable opportunity to examine the impact of various precursors and
aqueous-phase processes on aerosol NOC formation.
Sample collection was carried out simultaneously in three cities from 16
December 2017 to 14 January 2018. $PM_{2.5}$ samples were collected every two or three
days with a duration of 24 hours onto prebaked (450°C for ~10 hours) quartz fiber





filters (Pallflex, Pall Corporation, USA) using a high-volume air sampler (Series 2031,
Laoying, China). One blank sample was collected at each sampling site. A total of 39
samples were collected, all of which were stored at −30°C. Meteorological data (e.g.,
temperature, relative humidity (RH) and wind speed) together with concentrations of
various pollutants (e.g., $SO_2$ and $NO_2$) were obtained from nearby environmental
stations. The sampling periods were classified as either "clean" or "haze" based on
whether the daily average concentration of $PM_{2.5}$ was below or above 75 μg m$^{-3}$
(Zhang and Cao, 2015; Xu et al., 2024).

**2.2. Chemical analysis**

The extraction and analysis methods for NOCs were consistent with those

described in our recent publication (Ma et al., 2024). Briefly, a portion of each filter
was extracted with methanol (LC-MS grade, CNW Technologies Ltd.) using
sonication in an ice bath (~4°C). The extracts were filtered through a 0.22 μm
polytetrafluoroethylene syringe filter and then concentrated to 300 μL under a gentle
stream of nitrogen gas. The concentrated extracts underwent composition analysis via
an ultra-performance liquid chromatography coupled with quadrupole time-of-flight
mass spectrometry equipped with an electrospray ionization (ESI) source (UPLC-ESI-
QToFMS, Waters Acquity Xevo G2-XS) (Wang et al., 2021c; Ma et al., 2024). This
analysis was done in both ESI+ and ESI– modes. The organic compounds were



separated on a $C_{18}$ column (2.1 × 100 mm, 1.7 µm particle size, Waters) with an 18-
minute gradient elution. The mobile phases comprised ultrapure water with 0.1%
formic acid (A) and methanol with 0.1% formic acid (B). Gradient elution was
conducted according to the following protocol: 1% B was held for 1.5 minutes,
followed by an increase to 54% B over a period of 6.5 minutes. Thereafter, the B was
increased to 95% over a period of 3 minutes. After reaching 100% B in one minute,
this state was maintained for 3 minutes. Finally, the concentration was returned to 1%
B in 0.5 minutes and held for 2.5 minutes. Due to uncertainties in ionization
efficiencies for different compounds (Ditto et al., 2022; Yang et al., 2023), an
intercomparison (mainly compared among samples within this study) of compound
relative abundance was conducted without accounting for differences in ionization
efficiency in the present study. This consideration was consistent with previous
studies (Xu et al., 2023; Jiang et al., 2022; Ma et al., 2024).

Another filter portion was ultrasonically extracted using Milli-Q water (~4°C ice

bath) to analyze the concentrations of inorganic ions and organic acids. The inorganic
ions (e.g., $NO_3^-$, $SO_4^{2-}$, $Cl^-$, $Ca^{2+}$, $Mg^{2+}$, $Na^+$, and $NH_4^+$) and organic acids (e.g.,
formic acid, acetic acid, oxalic acid, succinic acid, glutaric acid, and methanesulfonic
acid) were quantified using an ion chromatograph system (Dionex Aquion, Thermo
Scientific, USA) as described previously (Xu et al., 2022b; Yang et al., 2024b).



**2.3. Compound categorization and precursor identification**


The identified molecular formulas via UPLC-ESI-QToFMS were categorized

into different compound classes based on their elemental compositions, which
included CHO−, CHON−, CHONS−, and CHOS− in ESI− mode and CHO+, CHON+,
and CHN+ in ESI+ mode (Ma et al., 2024). Here, we mainly focus on NOCs (i.e.,
CHN+, CHON+, and CHON− compounds) (Ma et al., 2024; Jiang et al., 2022; Wang
et al., 2017). The carbon oxidation state ($OS_C$) and double bond equivalent (DBE)
were calculated to indicate the oxidation level and unsaturation degree of the organics,
respectively (**Sect. S1**) (Kroll et al., 2011; Ma et al., 2024). Additionally, the modified
aromaticity index ($AI_{mod}$) and aromaticity equivalent ($X_C$) were used to evaluate
aromaticity of organics (Koch and Dittmar, 2006), as detailed in **Sect. S1**.

The potential precursors of NOCs were identified based on the methodology

outlined in previous studies (Nie et al., 2022; Guo et al., 2022; Jiang et al., 2023).
The classification of CHON+ and CHON− compounds was refined into following
categories, including aliphatics-, heterocyclics-, and aromatics-derived Re-NOCs
and isoprene-, monoterpenes-, aliphatics-, and aromatics-derived Ox-NOCs.
Moreover, CHN+ compounds were classified into aliphatic, monoaromatic, and
polyaromatic CHN+ compound (Wang et al., 2021b; Yassine et al., 2014). A detailed
description of the revised workflow for classifying NOCs according to potential
precursors was provided in **Sect. S2** and **Fig. S2**.




**2.4. Classification of potential pathways for NOC formation**

To identify potential aqueous-phase processes for aerosol NOC formation, we
screened precursor-product pairs from the organic compounds that have been detected
(Su et al., 2021; Xu et al., 2023; Jiang et al., 2023). The reaction pathways of Re-
NOCs (mainly CHON+ compounds in this study) were refined into the following
categories, including condensation (cond_N), hydrolysis (hy_N), dehydration (de_N),
cond_hy_N (involving cond_N and hy_N), cond_de_N (involving cond_N and de_N),
hy_de_N (involving hy_N and de_N), cond_hy_de_N (involving cond_N, hy_N and
de_N) and unknown_N (unknown processes) formation pathways (**Fig. S3 and Table
S4**) (Sun et al., 2024; Abudumutailifu et al., 2024; Laskin et al., 2014; Liu et al.,
2023c). Another significant class of Re-NOCs is the CHN+ compounds. Their
potential formation mechanisms include cond_N, de_N, cond_de_N, and other
unidentified (unknown_N) pathways (**Fig. S4 and Table S4**) (Abudumutailifu et al.,
2024; Laskin et al., 2014; Liu et al., 2023c). In addition, the reaction pathways of Ox-
NOCs (mainly CHON− compounds in this study) were refined into the following
categories, including ox_N, hy_N, ox_hy_N (involving ox_N and hy_N), and other
unidentified (unknown_N) pathways (Jiang et al., 2023; Su et al., 2021) (**Fig. S5 and
Table S4**). A detailed overview of the methodologies employed to discern potential
NOC formation pathways was shown in **Sect. S3**, **Table S4,** and **Figs. S3–S5**.





It is important to acknowledge the potential limitations in the categorization
methodology of NOC formation pathways described above. This is because the
approach applied here and in previous studies (Jiang et al., 2023; Su et al., 2021) may
classify NOCs from primary emissions as products of secondary aqueous-phase
reactions. Accordingly, our results should be regarded as indicating a maximal
potential (or an upper limit) for NOC generation from aqueous-phase reactions. In this
study, NOCs produced from the reaction pathways identified by the abovementioned
classification methodology can explain 76% of CHON+ compounds, 61% of CHN+
compounds, and 65% of CHON− compounds. Thus, the classification of potential
pathways for NOC formation was representative, at least in this study.

**2.5. More parameter calculations and data analysis**
A thermodynamic model (ISORROPIA-II) was used to estimate the ALW
concentration and pH value, as described in previous studies (Xu et al., 2020b; Xu et
al., 2023; Xu et al., 2022c). Ambient hydroxyl radical (•OH) concentrations were
predicted using empirical formulas proposed by Ehhalt and Rohrer (Ehhalt and Rohrer,
2000), which was reported in detail in our previous field observations (Liu et al.,
2023a; Lin et al., 2023). The ventilation coefficient (VC) is an indicator of the
potential for atmospheric dilution of pollutants, which was calculated by multiplying
the wind speed by the planetary boundary layer height (PBLH) (Gani et al., 2019) .



Non-metric multidimensional scaling (NMDS) was employed to visualize the
distributions of NOCs (CHON+, CHN+, and CHON– compounds) in two dimensions,
based on Bray-Curtis distances (Chao et al., 2006). The stress values ranged from
0.03 to 0.11 (less than 0.2, **Table S5**) in our analysis, indicating that the differences
among samples can be well represented in the two-dimensional pattern. To further
assess the influence of anthropogenic emissions and aqueous-phase processes on the
distribution of NOCs, the envfit function in the R package Vegan (Oksanen et al.,
2007) was utilized. Furthermore, the Spearman rank correlation, a non-parametric
measure with less sensitivity to outliers and independent of data distribution
assumptions, was employed to examine the association patterns between NOCs and
the parameters related to anthropogenic emissions and aqueous-phase processes
(Kellerman et al., 2014).

**3. Results and discussion**
**3.1. Overview of pollution and aerosol NOC characteristics in different cities**

**Figure 1a–c** and **Table S1** show the variations in major gaseous pollutants, $PM_{2.5}$

and its major compositions, as well as meteorological parameters in three Chinese
cities with different energy consumptions during winter. The average $PM_{2.5}$
concentration in HEB was $90.6 \pm 62.4$ µg m$^{-3}$, which was significantly higher than
that observed in BJ ($52.7 \pm 51.4$ µg m$^{-3}$) and HZ ($69.1 \pm 29.6$ µg m$^{-3}$). Similarly, the



concentrations of $SO_2$ and nss-$Cl^-$ were higher in HEB than in BJ and HZ. In addition,
a lower $NO_3^-/SO_4^{2-}$ mass ratio (**Table S1**) was found in HEB. $SO_2$ and nss-$Cl^-$ have
been suggested to be typical pollutants emitted from coal combustion during winter in
cities (Zhao and Sun, 1986; Streets and Waldhoff, 2000). The low $NO_3^-/SO_4^{2-}$ mass
ratio can indicate a predominance of stationary sources (e.g., coal combustion) (Wang
et al., 2006; Arimoto et al., 1996; Xiao and Liu, 2004). These results suggest that coal
combustion during the winter heating season in HEB may significantly contributed to
severe $PM_{2.5}$ pollution. This consideration can also be supported by the highest coal
consumption in HEB in 2017–2018 (**Fig. 1d**). Due to the large-scale use of clean
energy (i.e., natural gas) in BJ (**Fig. 1e**), the coal consumption in BJ was the lowest
(**Fig. 1d**). This results in the lowest pollutant levels in BJ. From clean to polluted days,
HEB and BJ showed larger increases in pollutant levels (e.g., $PM_{2.5}$, $SO_2$, and CO),
followed by HZ. Thus, the release of pollutants caused by the use of fossil fuels for
centralized heating in winter (only occurred in HEB and BJ) is undoubtedly one of the
important factors contributing to the generation of haze in HEB and BJ.





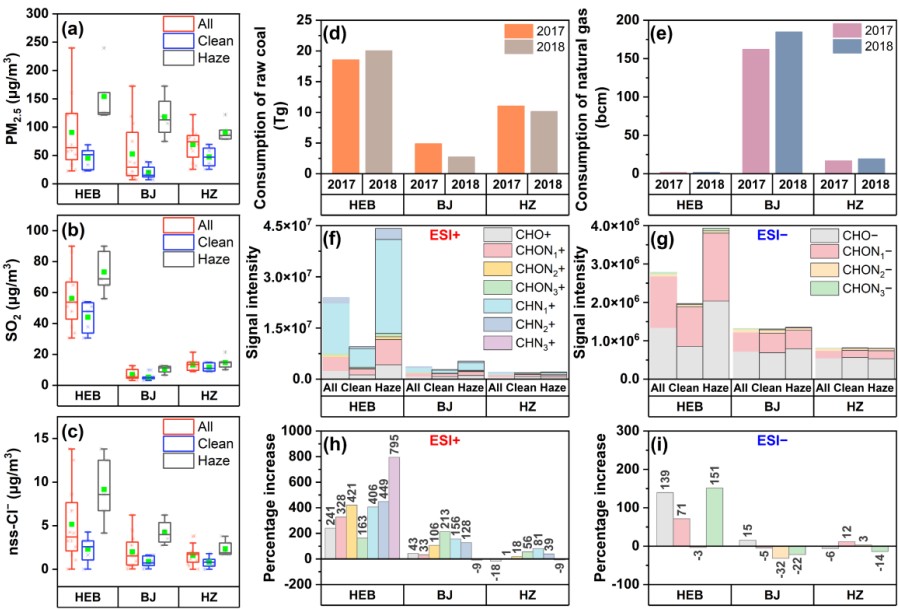

**Figure 1.** Box and whisker plots showing variations in the concentration of (**a**) PM$_{2.5}$,

(**b**) SO$_2$, and (**c**) nss-Cl$^-$in all (gray), clean (blue), and haze (red) periods in different

cities. Each box encompasses the 25th–75th percentiles. Whiskers are the 5th and

95th percentiles. The green squares and solid lines inside boxes indicate the mean and

median value. The consumption of (**d**) raw coal and (**e**) natural gas in 2017 and 2018

in different cities was obtained from the local statistical yearbooks. Average

distributions in the signal intensity of species detected in PM$_{2.5}$ collected during

different winter periods in different cities in (**f**) ESI+ and (**g**) ESI− modes. Percentage

variations in the signal intensity of each subgroup from clean to haze periods in

different cities in (**h**) ESI+ and (**i**) ESI− modes.



**Figure 1f** and **g** show the average signal intensity distributions of organic
compounds detected in $PM_{2.5}$ across sampling periods in different cities. The detailed
mass spectra of organic compounds detected in ESI+ and ESI− were shown in **Fig. S6**.
$CHN_1+$ ($n = 437−448$) compounds were the main CHN molecules measured in ESI+
mode in all cities (**Fig. 1f** and **Table S6**), the signal intensity of which accounted for
over 77% of the total $CHN_{1−3}+$ signal intensity. Similarly, $CHON_1+$ compounds ($n =$
$398−421$) dominated in $CHON_{1−3}+$ molecules, with a higher signal intensity than
$CHON_{2−3}+$ (**Fig. 1f** and **Table S6**). The high abundances of $CHN_1+$ and $CHON_1+$
compounds in NOCs were similar to previous reports about the NOC characteristics
of urban aerosols (He et al., 2024; Abudumutailifu et al., 2024). The signal intensity
fractions (40%−77%) of CHN+ compounds in total NOCs in these three cities were
higher than those observed (8.20%−17.47%) during winter in Ürümqi where the same
NOC analysis method was conducted (Ma et al., 2024). However, the signal intensity
fractions of CHON+ compounds in total NOCs were lower in these three cities
(23%−60%) than in Ürümqi (over 82.53%) (Ma et al., 2024). More frequent biomass
burning and relatively dry climate in Ürümqi (northwest China) (Ma et al., 2024) may
result in different sources and formation processes of NOCs compared to this study.
The signal intensity of these NOCs detected in ESI+ mode varied across cities, with
the highest CHN+ and CHON+ signal intensities in HEB, followed by BJ and HZ.
Moreover, we found that the total signal intensities of CHN+ and CHON+ compounds





increased by 382% in HEB from clean to haze periods, followed by increase of 102%
in BJ and increase of 31% in HZ (**Fig. 1h** and **Table S5**). This variation pattern of
CHN+ and CHON+ compounds from clean to haze periods was similar to that of the
pollutants mentioned previously (**Fig. 1a–c**). Given the high sensitivity of ESI+ mode
to protonatable species, reduced species (e.g., amine- and amide-like compounds)
were expected to predominate the NOCs (Han et al., 2023; Wang et al., 2018), the
formation of which was highly related to precursor emission level, aerosol acidity, and
ALW concentrations (Kuwata and Martin, 2012; Vione et al., 2005; Yang et al., 2024a;
Xu et al., 2020b). Thus, these results suggest that there may be significant differences
in the sources, precursor emission intensity, or main formation pathways of NOCs in
different energy consuming cities.
The number of NOCs identified in ESI− (296–301 molecules excluding sulfur-
containing compounds, **Table S7**) was found to be lower than that observed in ESI+
(1346–1361) (**Table S6**). This finding was similar to previous observations about the
NOCs of urban organic aerosols in Beijing, Mainz, Changchun, Guangzhou, and
Shanghai (Wang et al., 2021b; Wen et al., 2023; Wang et al., 2018). $CHON_1$−
compounds were the main NOC molecules in ESI− mode in all cities (**Fig. 1g** and
**Table S7**). The average signal intensity of CHON− compounds was highest in HEB,
followed by BJ and HZ. Moreover, the outbreak of $CHON_{1-3}$ signal intensity during
polluted periods was found in HEB, whereas insignificant increases occurred in BJ



and HZ (**Fig. 1i**). Deprotonated NOCs with oxidized nitrogen-functional groups, such
as nitro (–$NO_2$) or nitrooxy (–$ONO_2$) groups, are more sensitive to the ESI− mode
(Wang et al., 2017; Jiang et al., 2023; Yuan et al., 2023). Clearly, the formation of
aerosol CHON− compounds was largely dependent on atmospheric oxidation capacity
and gas- and aqueous-phase reactions (Ng et al., 2017; Shi et al., 2020; Shi et al.,
2023). Thus, the differences in CHON− compound abundance in different polluted
periods and cities together with the spatiotemporal changes in CHN+ and CHON+
abundances mentioned previously were likely attributed to variations in sources,
mechanisms, or key influencing factors of NOC formation in these three cities, which
will be further discussed in the following sections.

**3.2. Potential precursors of aerosol NOCs in different cities**

**Figure 2** presents the average signal intensity percentage and signal intensity

distributions of different NOCs from various precursors in different cities during
winter. Aromatics-, heterocyclics-, and aliphatics-derived Re-NOCs together
accounted for more than 74% (74%–79%) of the total signal intensity of CHON+
compounds in the three study cities (**Fig. 2a–c** and **Table S8**). Specifically, the
proportion of the aromatics-derived CHON+ signal intensity in the total CHON+
signal intensity was much higher in HEB (73%) than in BJ (33%), with the lowest
proportion observed in HZ (23%) (**Fig. 2a–c**). Furthermore, we observed that



aromatic CHN+ compounds (mono- and poly-aromatics) dominated the total CHN+
compounds in both number and abundance in all investigated cities (**Table S9** and **Fig.**
**2d–f**). The average signal intensity percentage and signal intensity of aromatic CHN+
compounds were also highest in HEB (**Fig. 2d–f** and **k**). The calculated $AI_{mod}$ values
for CHON+ and CHN+ compounds were higher in HEB than in BJ and HZ (**Table**
**S10**), which further indicated a higher aromaticity of these NOCs in HEB. It has been
suggested that coal combustion can release a large amount of aromatic compounds
(Zhang et al., 2023a), which potentially increased NOC aromaticity (Yuan et al.,
2023). Thus, the higher signal proportion of aromatics-derived Re-NOCs in HEB can
be explained by the higher coal combustion emissions during winter. In contrast, the
use of clean energy during the central heating season in BJ and the reduced emissions
in HZ without central heating weakened the formation of aerosol aromatic NOCs.

CHON− compounds were also primarily dominated by aromatics-derived Ox-

NOCs in all three cities, accounting for more than 73% (73%–90%) of the total signal
intensity of CHON− compounds, on average (**Fig. 3g–i**). This finding was consistent
with field observations conducted in other Chinese cities such as Shanghai,
Changchun, Guangzhou, and Wangdu during winter (Wang et al., 2021b; Jiang et al.,
2023). The abundance of aromatics-derived Ox-CHON− compounds and the $AI_{mod}$
value of CHON− were highest in HEB and decreased sequentially in BJ and HZ (**Fig.**
**2h** and **Table S10**), further indicating our previous consideration that coal combustion



heating in HEB can lead to higher NOC pollution. It is worth noting that the
percentage of total signal intensity of Ox-NOCs with biogenic VOCs (BVOCs) as
precursors was less than 3% (**Fig. 2g–i** and **Table S8**). This can be partly supported by
the previous observations showing that anthropogenic VOCs (AVOCs) were the main
contributors to the formation of Ox-NOCs (e.g., organic nitrates) in urban areas in
China (Wang et al., 2021b; Jiang et al., 2023). The overall results suggest the
significant role of AVOCs in the formation of NOCs in all investigated cites,
particularly in HEB.

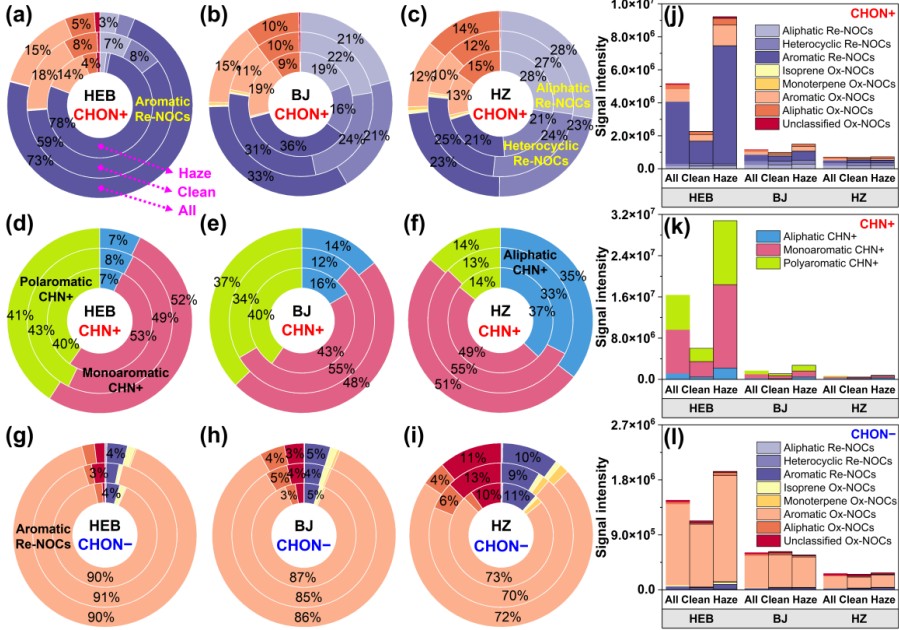

**Figure 2.** Average percentage distributions of signal intensities for (**a–c**) CHON+,
(**d–f**) CHN+, and (**g–i**) CHON− compounds from various sources in PM$_{2.5}$ collected
from different cities during winter. Average signal intensity distributions for (**j**)





CHON+, (**k**) CHN+, and (**l**) CHON− compounds from various sources in $PM_{2.5}$
collected from different cities during winter.

From clean to haze periods, the signal intensities of all aromatics-derived CHON
compounds increased significantly in HEB (**Figs. 2a, j, g, l** and **S7**). In contrast, the
signal intensities of aromatics-derived CHON compounds in BJ and HZ showed an
insignificant increase during haze periods. In addition, the average values of $O/C_w$ and
$OS_{Cw}$ for CHON+ and CHON− compounds were higher in HEB than in BJ (second
highest) and HZ, and their increases from clean to haze periods were also greater in
HEB (**Table S10**). This indicates that aerosol NOCs in HEB were more oxidized
aromatics, particularly during haze. The average $N/C_w$ ratios of CHON+ and CHON−
compounds in HEB (0.13 and 0.15, respectively) (**Table S10**) were higher than those
of CHON+ (0.079) and CHON− (0.07) compounds in aerosols directly emitted from
coal combustion (Song et al., 2022; Song et al., 2018). The $N/C_w$ ratios were also
higher in HEB than in BJ and HZ and increased during hazy days (0.13 for CHON+
and 0.16 for CHON− in hazy days in HEB). It has been suggested that the $N/C_w$ ratio
of CHON− compounds tended to increase (from 0.109 to 0.119) after aging of fuel
combustion-derived aerosols (Zhao et al., 2022a). Thus, these results, combined with
previous analysis of potential precursors for NOCs, suggest that anthropogenic
precursor emissions and their atmospheric transformation to form CHON compounds





were stronger in HEB than in BJ and HZ. Moreover, considering that the emission
intensity of precursors during clean and hazy days may not significantly change,
secondary processes may significantly promote the formation of NOCs in HEB during
hazy days (the most significant increase in NOC abundance). However, this
promoting effect during hazy days was insignificant in BJ and HZ (less increase in
NOC abundance).

**3.3. Main factors influencing aerosol NOC formation in different cities**
The Spearman correlation analysis between various parameters and NOCs was
conducted to investigate the key factors influencing NOC molecular distributions **(Fig.**
**3** and **Figs. S8−S12**). The peak intensity of most CHON+ compounds (mainly
aromatics, as mentioned previously) showed a strong correlation ($P < 0.01$) with the
concentrations of combustion source-related tracers (e.g., $SO_2$, nss-$Cl^-$, nss-$K^+$, CO,
and $NO_2$) (Zhao and Sun, 1986; Streets and Waldhoff, 2000; Shen et al., 2009; Zhang
et al., 2011; Mafusire et al., 2016; Liu et al., 2019; Zhang et al., 2021a; Wang et al.,
2020) in HEB (**Figs. 3a** and **S8a–d**). Although there was a significant correlation ($P <$
$0.05$) between most CHON+ compounds and those combustion source indicators in
BJ, the strength of this correlation was weaker in BJ than in HEB (**Figs. 3e** and **S8f–i**).
However, similar significant correlations between them were not observed in HZ
(**Figs. 3i** and **S8k–n**). Thus, the greatest contribution of anthropogenic activities to the



formation of CHON+ compounds in winter was in HEB (central heating with coal),
followed by BJ (central heating with coal and natural gas) and HZ (without central
heating). Most of CHN+ and CHON− compounds showed a similar spatial response
pattern to those anthropogenic activities (**Figs. S9** and **S10**). These results are
consistent with the previous analysis of NOC precursors (**Fig. 2**), which concluded
that the intensity of anthropogenic pollutant emissions in different energy consuming
cities was an important factor affecting the formation of NOC and causing spatial
differences in NOC abundance.

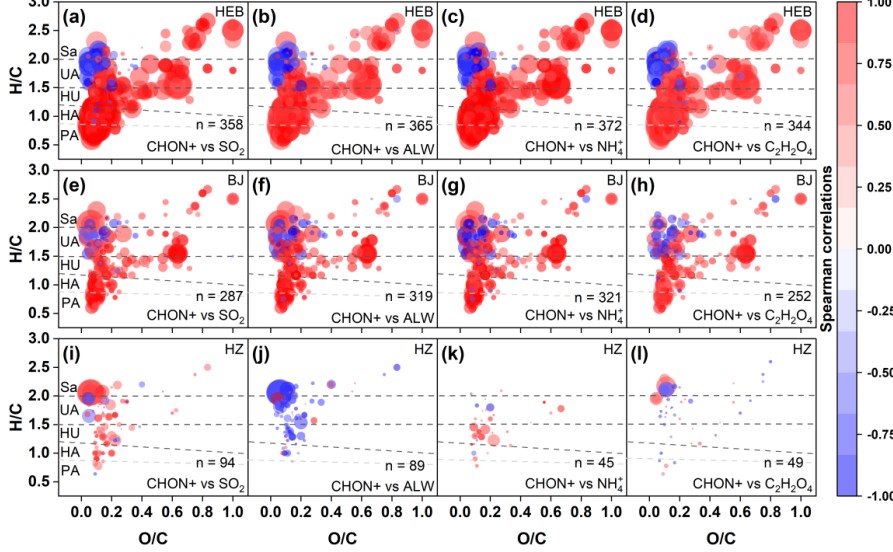


**Figure 3.** Spearman rank correlation coefficients (with $P < 0.01$ in HEB and $P < 0.05$
in BJ and HZ) of individual CHON+ molecules with selected parameters in (**a–d**)
HEB, (**e–h**) BJ and (**i–l**) HZ. The color scale indicates Spearman correlations between
the intensity of individual CHON+ molecules and each parameter. The symbol *n* in



the bottom right corner of each panel indicates the number of molecular formulas
significantly correlated with the variables. The subgroups in the panels include
polycyclic aromatic-like (PA), highly aromatic-like (HA), highly unsaturated-like
(HU), unsaturated aliphatic-like (UA), and saturated-like (Sa) compounds.

Furthermore, we found that the peak intensities of most CHON+, CHN+, and

CHON− compounds (mainly aromatics) were significantly correlated ($P < 0.01$) with
the concentrations of ALW, $NH_4^+$, oxalic acid, and $SO_4^{2-}$ (**Figs. 3b–d**, **S8e**, and
**S11–S12**) in HEB. The correlations between these NOCs and parameters weakened in
BJ and disappeared in HZ (**Figs. 3**, **S8**, and **S11–12**). It is generally accepted that
$SO_4^{2-}$, $NH_4^+$, and $NO_3^-$ in fine aerosols are primarily formed through secondary
processes (Gao et al., 2021; Wang et al., 2021d). $NH_4^+$ can serve as a key reactant in
the formation of aerosol NOCs (e.g., "carbonyl-to-imine" transformation) in the
aqueous-phase (Laskin et al., 2014; Lee et al., 2013; Li et al., 2019b). Previous studies
have identified oxalic acid ($C_2H_2O_4$) as a tracer for aqueous-phase SOA (Xu et al.,
2022a; Carlton and Turpin, 2013). Additionally, numerous laboratory and field
observational studies have shown that ALW can promote the formation of NOCs (Lv
et al., 2022; Liu et al., 2023b; Jimenez et al., 2022; Jiang et al., 2023). Thus, these
results indicate that aqueous-phase processes can significantly promote the formation
of NOCs in HEB, however, as the precursor emission intensity gradually decreased in

1

BJ and HZ, this aqueous-phase promoting effect also decreased.

The NMDS analysis between various parameters and NOCs was conducted to

further investigate the variations in key factors affecting the formation of NOCs from
clean to haze days (**Fig. 4**). The formation of CHON+, CHN+, and CHON−
compounds with higher $AI_{mod}$ values (mainly aromatics, as mentioned previously)
during haze days in HEB and BJ were closely associated with the factors indicating
anthropogenic precursor emissions and aqueous-phase reaction processes. In contrast,
the level of oxidants (i.e., $O_3$ and •OH) played a more important role during clean
days in HEB and BJ, driving more highly saturated NOC formation during clean days
(**Fig. 4**). A reasonable explanation for this is that the solar radiation and •OH levels on
polluted days were lower than those on clean days (**Table S1**). The impacts of various
factors on the formation of aerosol NOCs showed a weak discrimination between
haze and clean days in HZ (**Fig. 4c, f** and **i**). Laboratory studies have shown that
reactive components (e.g., •OH and $H_2O_2$) in the aqueous phase can continuously
convert low-solubility organics to form aqueous phase SOA (Chen et al., 2008; Huang
et al., 2011). Field observations also suggested that precursors (most of them are
aromatic compounds) released from the combustion of fossil fuels significantly
contributed to the aqueous SOA formation (> 50% total molecules) (Xu et al., 2022a)
through the rapid aqueous-phase conversion of primary organic aerosol (POA) to
SOA at high RH (Wang et al., 2021a). This implies that higher precursor abundance



can drive more aerosol NOC formation via aqueous-phase processes. As mentioned
previously, the emission intensity of precursors decreased sequentially from HEB to
BJ and then to HZ. Moreover, the ALW concentrations were much higher on polluted
days than on clean days in three investigated cities. The rising ALW during the
pollution period and the quiescent steady state of the atmosphere favored the
formation of SOA from anthropogenic emission precursors (Guo et al., 2014; He et al.,
2018). Thus, the above discussion can suggest that the spatial differences in precursor
emission intensity (higher in HEB) and enhancement of aqueous-phase processes in
polluted days were the main factors leading to the differences in the proportion
(higher in HEB) of increase in NOC abundance from clean days to polluted days in
three different energy consuming cities. In addition, the increased VC value (**Table**
**S1**) in clean days (beneficial for the diffusion of pollutants) (Gani et al., 2019) was
also an important factor limiting the abundance of NOCs (**Fig. 4**), resulting in a lower
NOC abundance on clean days compared to polluted days (**Fig. 1**).



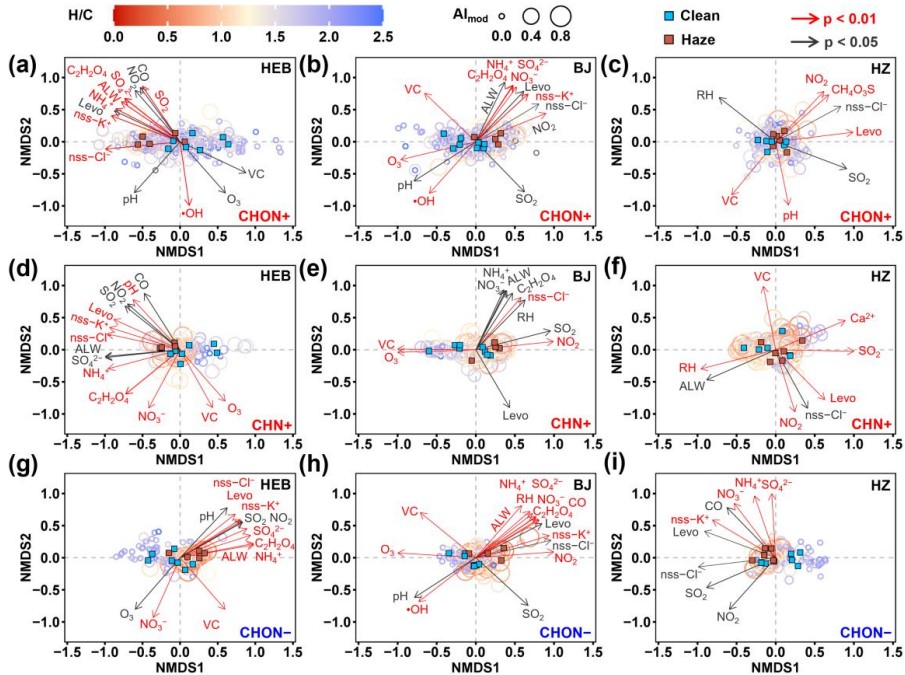

**Figure 4.** Nonmetric multidimensional scaling of (**a–c**) CHON+, (**d–f**) CHN+, and

(**g–i**) CHON− compounds from organic aerosol in different cities. The color and size

of the circle indicate the H/C ratio and $AI_{mod}$ value of compounds, respectively.

Significant relationships between the variables and ordination (999 permutations) are

indicated by $p < 0.05$ (grey) and $p < 0.01$ (red). Insignificant correlations are not

shown. The scores of the samples collected during clean and haze periods were shown

as blue and brown squares, respectively.

As mentioned above, the aerosol NOCs of HZ were less affected by

anthropogenic pollutants emitted from coal and natural gas combustion compared to



HEB and BJ with centralized heating. Interestingly, we found that the molecular
distributions of most aromatic CHON+ compounds in HZ were not only influenced
by some anthropogenic pollutants (e.g., $SO_2$ and $NO_2$), but also by methanesulfonic
acid ($CH_4O_3S$) (**Fig. 4c**). Moreover, neither CHN+ nor CHON+ exhibited significant
correlations with factors related to secondary processes in HZ (**Fig. 4c** and **f**).
Methanesulfonic acid has been suggested to be a tracer for ocean aerosols (Ayers and
Gras, 1991; Suess et al., 2019). These results suggest that aerosol CHON+
compounds in HZ may be influenced by long-distance transport air masses originating
from the ocean. This consideration can be also supported by the fact that only HZ was
affected by air masses originated from the ocean (**Fig. S13**). Thus, marine emissions
may be an important contributor to aerosol NOCs in HZ, which was significantly
different from the cases of HEB and BJ where aromatic pollutants from fossil fuel
combustion and aqueous-phase processes control the composition and abundance of
aerosol NOCs.

**3.4. Potential formation mechanisms of aerosol NOCs in cities with different**
**energy consumption**

**Figure 5** shows the average signal intensity percentage and signal intensity

distributions of NOCs formed by different aqueous-phase processes (**Table S4** and
**Figs. S3–S5**) in different cities during winter. The identification of specific reaction



pathways was detailed in **Figs. S3–S5** and **Sect. S3**. During the entire study period,
the cond_N, cond_hy_N, and cond_de_N pathways together accounted for more than
68% (68%−74%) of the total signal intensity of CHON+ compounds in the three cities
(**Fig. 5a–c** and **Table S11**). Specifically, the formation of CHON+ compounds was
mainly dominated by the cond_N and cond_hy_N pathways in HEB, with less impact
from the cond_de_N pathway (**Fig. 5a**). However, CHON+ compounds derived from
the cond_de_N pathway showed a much higher proportion in BJ and HZ than in HEB
(**Fig. 5b** and **c**). The cond_de_N pathway involves both condensation and dehydration
processes (**Table S4 and Fig. S3**). It has been suggested that higher temperatures can
facilitate the dehydration of amides into nitriles (Mekki-Berrada et al., 2013). The
temperatures in BJ and HZ were higher than those in HEB (**Table S1**), which may
partly explain the higher signal proportion of CHON+ compounds formed through the
cond_hy_N pathway in BJ and HZ than in HEB. Furthermore, the higher signal
proportions of CHN+ compounds formed through the de_N pathway in BJ (6%) and
HZ (11%) than in HEB (2%) may also be associated with this temperature-induced
dehydration mechanism (**Fig. 5d–f and Table S12**). For CHN+ compounds, the
cond_de_N process dominated their formation (**Fig. 5d–f**). In general, the cond_N,
cond_hy_N, and cond_de_N processes contributed most significantly to the formation
of Re-NOCs in HEB, followed BJ and HZ.



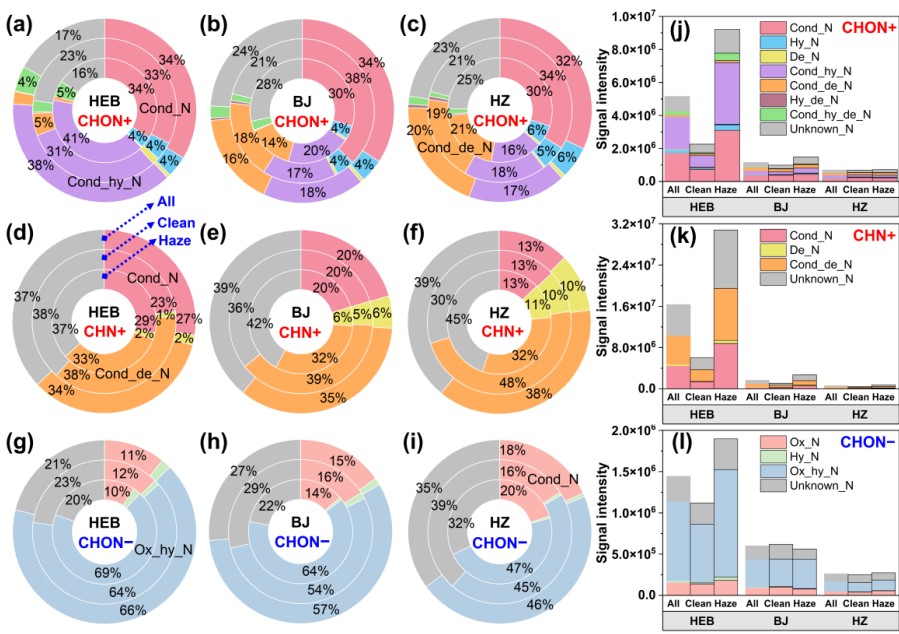

**Figure 5.** Average percentage distributions of signal intensities for aerosol (**a–c**) CHON+, (**d–f**) CHN+, and (**g–i**) CHON− compounds from various reaction pathways in different cities during winter. Average signal intensity distributions for aerosol (**j**) CHON+, (**k**) CHN+, and (**l**) CHON− compounds from various reaction pathways in different cities during winter.

A typical mechanism for Re-NOC formation is the aqueous-phase reactions between carbonyl compounds and $NH_4^+$ (or $NH_3$) (Abudumutailifu et al., 2024; Laskin et al., 2014; Li et al., 2019b; Liu et al., 2023b; Wang et al., 2024). If this mechanism is simplified as a second-order reaction (i.e., [Precursor] + [$NH_3$ and $NH_4^+$] ↔ [Re-NOCs]), the production of Re-NOCs is expected to be proportional to the





abundances of precursor and $NH_4^+$ (Yang et al., 2023; Lin et al., 2023). Indeed, the
signal intensities of the Re-CHON+ and Re-CHN+ compounds were significantly
positively correlated with the signal intensities of their CHO precursors (identified
using the precursor-product pairs theory, **Figs. S3** and **S4**) and $NH_4^+$ concentration in
HEB (**Fig. 6a, b, d and e**). This correlation gradually weakened from BJ to HZ (**Fig.**
**6a, b, d** and **e**). As previously discussed, differences in energy consumption patterns
resulted in the highest levels of anthropogenic aromatic compound emissions in HEB
during the winter, followed by BJ, with the lowest levels in HZ (**Figs. 2** and **S14**).
Thus, the signal intensities of CHON+ and CHN+ compounds from cond_N,
cond_de_N, and cond_hy_N processes were higher in HEB than in BJ and lowest in
HZ (**Fig. 5j** and **k**).

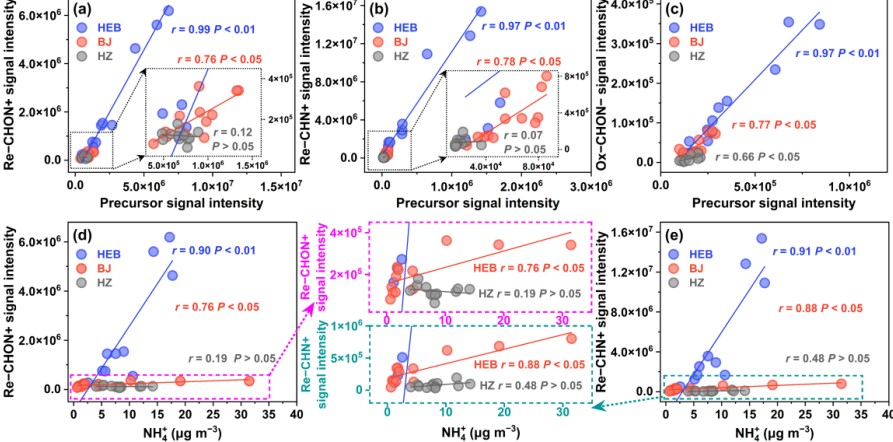


**Figure 6.** Signal intensity of (**a**) Re-CHON+, (**b**) Re-CHN+, and (**c**) Ox-CHON−
compounds as functions of signal intensity of precursors (CHO compounds). Signal
intensity of (**d**) Re-CHON+ and (**e**) Re-CHN+ compounds as functions of the



concentrations of $NH_4^+$.

Additionally, we noticed that the contribution of these aqueous-phase processes to
the formation of CHON+ and CHN+ compounds increased significantly from clean to
hazy days in HEB and BJ (**Fig. 5**). The increased ALW concentrations (**Table S1**) and
atmospheric stability during haze periods likely provided favorable conditions for the
precursors to undergo these aqueous-phase reactions, resulting in the formation of
NOCs. Clearly, high pollutant emission levels in HEB provided a greater potential to
convert precursors into more NOCs via the cond_N, cond_hy_N, and cond_de_N
processes during haze periods. Thus, the hazy days in the HEB showed the largest
increase in CHON+ and CHN+ compounds from the cond_N, cond_hy_N, and
cond_de_N processes (**Fig. 5j** and **k**). In contrast, the lower precursor emissions in
HZ without centralized heating policy were not sufficient to support the production of
large amounts of NOCs in the aqueous phase. These results also indicate that emission
reduction is the key to controlling aerosol NOC pollution.
CHON− compounds derived from the ox_hy_N and ox_N processes together
accounted for more than 64% (64%−71%) of the total signal intensity of CHON−
compounds in the three cities (**Fig. 5g–i, l** and **Table S13**). The signal intensity
proportions of CHON− compounds formed by the ox_hy_N process in these three
cities (ranging from 47% in HZ to 69% in HEB) were higher than that in Wangdu (<





20%) (Jiang et al., 2023). The observation study in Wangdu examined aerosol organic
components only in ESI− mode (Jiang et al., 2023), which may underestimate the
importance of the CHO+ compounds that could serve as precursors of Ox-NOCs. In
general, CHON− compounds formed through the ox_hy_N and ox_N processes
showed the highest abundance in HEB, followed by BJ and HZ (**Fig. 5j–i**). According
to a simplified reaction mechanism for the formation of Ox-NOCs via aqueous-phase
processes (i.e., [Precursor] + [Oxidants] ↔ [Ox-NOCs]) (Shi et al., 2023; Kroflič et
al., 2015; Vione et al., 2005), we can infer that Ox-NOCs production is proportional
to precursor levels when oxidants (e.g., $NO_2$ radical or $NO_2^+$) are in a steady state in
the atmosphere. Indeed, the signal intensities of the Ox-CHON− compounds were
significantly positively correlated with the signal intensities of their CHO precursors
identified using the precursor-product pairs theory in HEB (**Fig. 6c**). Moreover, this
correlation gradually weakened from BJ to HZ (**Fig. 6c**). Thus, the spatial differences
in the contribution of the ox_hy_N and ox_N processes to Ox-NOC production across
the three cities can also be explained by differences in precursor emission intensity, as
indicated by above mentioned Re-NOC formation.

**4. Conclusion**
The abundance, composition, potential precursors, and potential formation
mechanisms of NOCs in $PM_{2.5}$ in three Chinese cities with different energy



consumption types during the winter were systematically investigated. On average,
the total signal intensity of NOCs (i.e., CHN+, CHON+, and CHON− compounds)
was highest in HEB, followed by BJ. The lowest total NOC signal intensity was found
in HZ. According to the identification of potential precursors of NOCs, we found that
anthropogenic aromatic compounds were the main precursors of NOCs during winter
in HEB where mainly relies on coal for winter heating, with less impact from BVOCs.
Anthropogenic aromatic precursors were also identified to be important contributors
to NOC formation in BJ which uses natural gas and coal for winter heating, although
the contribution ratio was lower in BJ than in HEB. In contrast, the lowest aromatic
precursor levels occurred in HZ without winter heating policy. Furthermore, the
NMDS analysis supported that the impact of anthropogenic fossil fuel combustion on
NOC pollution gradually decreased from HEB to BJ and then to HZ.

The formation of CHON+ compounds was mainly associated with the cond_N,

cond_hy_N, and cond_de_N processes. The cond_N and cond_de_N processes
dominated the formation of CHN+ compounds. The production of CHON+ and
CHN+ compounds from the cond_N, cond_hy_N, and cond_de_N processes was
highest in HEB, followed by BJ and HZ. The ox_hy_N and ox_N processes
contributed significantly to CHON− compound formation, from which the highest
CHON− production occurred in HEB and the lowest in HZ. The spatial differences in
the contribution of different aqueous-phase processes to NOC production in the three



different cities can be attributed to differences in precursor emission intensity. In
particular, the contribution of these aqueous-phase processes to the formation of
CHON+ and CHN+ compounds showed the most significant increase from clean to
hazy days in HEB, followed by BJ. We concluded that high pollutant emission levels
can provide a greater potential to convert precursors to produce more NOCs via
aqueous-phase processes during haze periods. The above findings are summarized in
a diagram (**Fig. 7**).

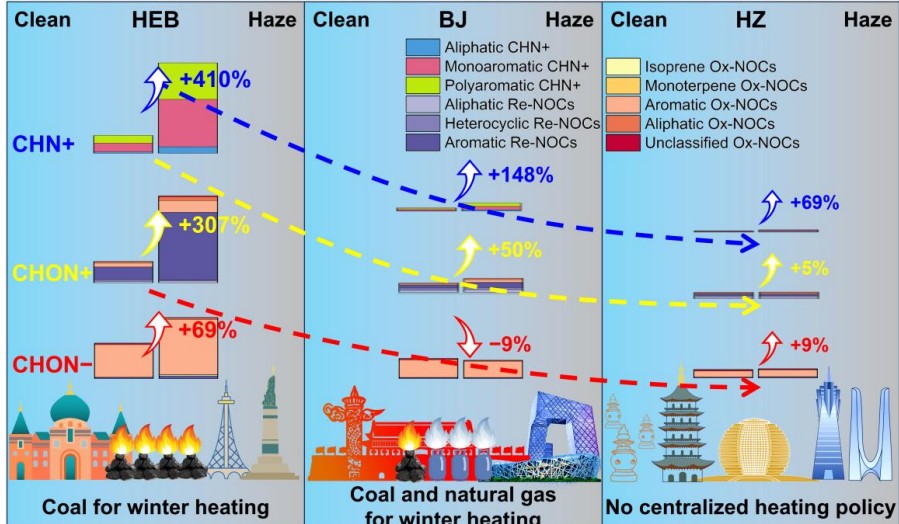


**Figure 7**. Conceptual illustration showing the characteristics of different NOCs from
the clean days to the haze days in different cities.
In general, the aerosol NOCs pollution during winter is closely linked to both the
intensity of precursor emissions and the efficiency of aqueous-phase processes in
converting these emissions into NOCs. The overall results highlight the importance of



emission reduction strategies in controlling aerosol NOCs pollution during winter.
Targeted reduction of precursor emissions, especially from coal combustion, could
significantly mitigate NOCs levels, thereby improving air quality and public health in
urban areas. Future research should focus on further elucidating the specific pathways
of aqueous-phase NOC formation and developing available models to predict NOC
dynamics under varying environmental conditions. Additionally, research into the
long-term effects of transitioning to cleaner energy sources on the reduction of NOC
pollution will be essential for guiding effective air quality management strategies.

**Data availability.** The data presented in this work are available upon request from the
corresponding authors.

**Competing interests.** The authors declare no conflicts of interest relevant to this
study.

**Supplement.** Details of parameter calculation, classification method for identifying
precursors of NOCs, classification of possible aqueous-phase processes NOCs based
on precursor-product pairs, thirteen tables (Tables S1−S13), and fourteen extensive
figures (Figures S1−S14) are provided in the Supplement.



**Author contributions.** YX designed the study. YJM, TY, LG, HX, and HWX
performed field measurements and sample collection; YJM performed chemical
analysis; YX and YJM performed data analysis; YJM and YX wrote the original
manuscript; and YX, YJM, and HYX reviewed and edited the manuscript.

**Financial support.** This study was kindly supported by the National Natural Science
Foundation of China through grant 42303081 (Y. Xu) and Shanghai "Science and
Technology Innovation Action Plan" Shanghai Sailing Program through grant
22YF1418700 (Y. Xu).

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
