# Peer review of "The Critical Role of Aqueous-Phase Processes in Aromatic-Derived Nitrogen-Containing Organic Aerosol Formation in Cities with Different Energy Consumption Patterns"

_EGUsphere, 2024_

## Author Comment (AC1)

**General.**

We would like to express our sincere appreciation to the editor and reviewers for their valuable feedback and constructive suggestions, which significantly improved the manuscript. We have carefully addressed all the reviewers' concerns and made the necessary revisions. Responses to specific comments raised by the reviewers are described below. All changes made in the revised manuscript are highlighted in red, and our detailed responses to the specific comments are presented below in blue.

**Comments of Referee #3 and our responses to them**

Comments:

*This study examines the formation and pollution of aerosol nitrogen-containing organic compounds (NOCs) in $PM_{2.5}$ across three Chinese cities during winter. The authors systematically compare the relative abundance, precursors, and formation mechanisms of various NOCs in cities with varying energy consumption patterns. The study emphasizes the significant role of anthropogenic aromatics and aqueous-phase processes in the occurrence of urban aerosol NOC pollution, particularly during haze days. The overall results provide valuable insights into the complex interactions between emissions, meteorological conditions, and aqueous-phase chemical processes in NOC formation. In general, this study advances our understanding of urban aerosol NOCs and will be of interest to Atmospheric Chemistry and Physics readers. The work is solid and well-written, and I recommend it for publication following minor revisions.*

Response: We are very grateful for your professional and thoughtful review of our manuscript. We have revised the manuscript to address the comments. Our responses to the specific comments and changes made in the manuscript are given below.

Specific comments:

1) *Line 69: "nitrogen oxide" change to "nitrogen oxides"*

Response: Thank you for pointing this out. The term "nitrogen oxide" has been corrected to "nitrogen oxides" in the revised manuscript (Lines 69-70).

2) *Line 80-82: I recommend rephrasing this sentence. Here is a suggestion: Several observational studies have found a positive correlation between aerosol NOC abundance and either aerosol liquid water (ALW) or relative humidity (RH).*

Response: Thank you for your suggestion. The revised sentence is clearer and more concise. The revision has been made in the revised manuscript (Lines 80-81).

3) *Line 87: 'large emissions of NOC precursors…'. A reference is recommended to be added here.*

Response: Thank you for your suggestion regarding Line 87. The relevant references have been added to support the statement in the revised manuscript (Lines 87-88).

4) *Line 169-172: "…which included CHO−, CHON−, CHONS−, and CHOS− in ESI− mode and CHO+, CHON+, and CHN+ in ESI+ mode". The use of "-" and "+" signs in the molecular formulas here needs clarification. It is unclear whether they refer to the detected ion forms or the molecular forms. This ambiguity should be addressed.*

Response: Thank you for your valuable comment regarding the clarification of the "−" and "+" signs. The "−" and "+" signs following the molecular formulas indicate the detection ion modes, corresponding to negative (ESI−) and positive (ESI+) electrospray ionization modes, respectively. To avoid ambiguity, we have added the following explanation in the manuscript as described below (Lines 176–179).

Lines 176–179: …Unless otherwise indicated, the molecular formulas presented in the manuscript refer to neutral molecules. The "−" and "+" symbols denote the detection ion modes, which correspond to ESI− and ESI+ modes, respectivel.

5) Line 222: "…proposed by Ehhalt and Rohrer (Ehhalt and Rohrer, 2000), which was reported…", the citation format of the reference is incorrect. Please correct it.

Response: The revision has been made in the revised manuscript (Line 233).

6) Line 227-231: The authors used non-metric multidimensional scaling (NMDS) for visualizing the distribution of NOCs. Have you considered alternative data visualization methods, such as principal component analysis (PCA)? If not, could you please clarify the specific advantages of using NMDS in this context?

Response: Thank you for your insightful comment regarding the use of non-metric multidimensional scaling (NMDS) for visualizing the distribution of NOCs.

We selected NMDS because it is well-suited for analyzing complex datasets with non-linear relationships and semi-quantitative data (Taguchi and Oono, 2004; Bialik et al., 2021). Unlike principal component analysis (PCA), which assumes linear relationships and is sensitive to data scaling (Bialik et al., 2021), NMDS does not rely on these assumptions and focuses on preserving the rank order of dissimilarities. This flexibility makes NMDS particularly suitable for our dataset, where the underlying relationships are likely non-linear. Thus, given these advantages, we decided to use NMDS for our analysis, as it better suited the nature of our data.

7) Section 2.4: Regarding the potential pathways for NOC formation, have the authors considered bimolecular reactions, such as oligomerization reactions, as potential mechanisms in the formation of NOCs?

Response: Thank you for your valuable comment regarding the potential pathways for NOC formation.

In this study, we primarily focused on the following reaction pathways: imination reaction, intramolecular N-heterocyclic reaction, hydrolysis reaction, dehydration reaction, oxidation reaction, and their mixed reactions. These pathways were able to explain 76% of CHON+ compounds, 61% of CHN+ compounds, and 65% of CHON− compounds, indicating that the classification of potential pathways for NOC formation in our study is representative.

While bimolecular reactions, such as oligomerization, could contribute to certain atmospheric processes, they were not included in this study. This is because the atomic variations involved in oligomerization are too complex to be effectively described using the "precursor-product pairs" approach employed here. To clarify, we have added the following statement to the revised manuscript (Lines 220-223).

Lines 220-223: …In particular, certain reaction pathways (e.g., oligomerization) were not included in this study due to the complexity of the atomic changes involved, which could not be effectively characterized using the "precursor-product pairs" approach...

We acknowledge the importance of considering a broader range of potential mechanisms, including oligomerization, for a more comprehensive understanding of NOC formation. We appreciate the reviewer's suggestion and will aim to explore this aspect in future studies.

8) *Could the authors clarify whether the higher NOC abundances observed during haze periods were mainly due to secondary processes, or could there also be a contribution from primary emissions?*

Response: Thank you for your insightful comment regarding the potential sources contributing to the higher NOC abundances observed during haze periods.

Indeed, precursor-product pairing approaches may classify NOCs from primary emissions as secondary aqueous-phase reaction products, as suggested by previous studies (Jiang et al., 2023; Su et al., 2021). Thus, our results should be interpreted as an upper limit for NOC formation associated with aqueous-phase processes. In addition, the NMDS and Spearman correlation analyses suggested that the secondary formation of NOCs was significantly enhanced during haze conditions, further supporting the importance of secondary processes in the formation of NOCs.

We also acknowledged the inherent limitations of this approach in the manuscript (Lines 215–220).

Lines 215–220: …It is important to acknowledge the potential limitations in the categorization methodology of NOC formation pathways described above. This is because the approach applied here and in previous studies (Jiang et al., 2023; Su et al., 2021) may classify NOCs as products of aqueous-phase reactions from primary emissions. Accordingly, our results can be regarded as a maximal potential (or an upper limit) for NOC generation from aqueous-phase reactions.

9) *The authors focus on the sources and formation mechanisms of NOCs. Could you briefly discuss the implications of your findings for air quality management strategies?*

Response: Thank you for your valuable comment. This study underscores that wintertime aerosol NOC pollution is strongly influenced by both the intensity of precursor emissions and the efficiency of aqueous-phase processes in transforming these precursors into NOCs. The findings emphasize the critical need for targeted

emission reduction strategies to control NOC pollution and improve air quality, particularly in coal-dependent cities like Harbin. Specifically, reducing emissions of anthropogenic aromatic precursors (the dominant contributors to NOC formation in Harbin) can significantly mitigate NOC pollution levels and, consequently, improve air quality..

In addition, our results highlight the importance of transitioning to clean energy to reduce precursor emissions. For example, the decreased gradient in NOC pollution from Harbin (coal-reliant) to Beijing (partially using natural gas) to Hangzhou (without centralized heating) demonstrates the effectiveness of clean energy policies in constraining NOC pollution. Transitioning away from coal combustion and toward cleaner energy, such as natural gas or renewable energy, could be a powerful tool in reducing urban NOC pollution and improving air quality.

Furthermore, the findings demonstrate that aqueous-phase processes, especially under haze conditions, amplify the production of NOCs. This highlights the importance of controlling precursor emissions during periods of high humidity and haze to limit secondary NOC formation.

Overall, these insights can improve air quality management by prioritizing (1) reductions in aromatic precursor emissions through stricter controls on coal combustion and other anthropogenic sources, (2) the promotion of cleaner energy transitions, and (3) targeted measures to address secondary formation processes under haze conditions. Future studies should explore the long-term benefits of these strategies and refine models for predicting NOC dynamics under different environmental scenarios to support more effective policymaking.

We have included a brief discussion of these implications in the revised manuscript (Lines 649-651 and 654-656):

Lines 649-656: …It is imperative to manage precursor emissions during hazy episodes in order to restrict the increased formation of secondary NOCs in conditions of high humidity. Moreover, …

Lines 649-656: …The transition to cleaner energy sources, as evidenced by the decreased gradient of NOC pollution from HEB to BJ to HZ, represents an effective pathway for the mitigation of NOC pollution.

10) *Figure 6: It seems that the city label in the inset of Figure 6 is incorrect. The red circle and line should indicate Beijing (BJ), not HEB.*

Response: The revision has been made (Line 568).

**At last, we deeply appreciate the time and effort you've spent in reviewing our manuscript.**

**Reference:**

Bialik, O. M., Jarochowska, E., and Grossowicz, M.: Ordination analysis in sedimentology, geochemistry and palaeoenvironment—Background, current trends and recommendations, The Depositional Record, 7, 541-563, https://doi.org/10.1002/dep2.161, 2021.

Jiang, H., Cai, J., Feng, X., Chen, Y., Wang, L., Jiang, B., Liao, Y., Li, J., Zhang, G., Mu, Y., and Chen, J.: Aqueous-Phase Reactions of Anthropogenic Emissions Lead to the High Chemodiversity of Atmospheric Nitrogen-Containing Compounds during the Haze Event, Environ. Sci. Technol., 57, 16500-16511, 10.1021/acs.est.3c06648, 2023.

Su, S., Xie, Q., Lang, Y., Cao, D., Xu, Y., Chen, J., Chen, S., Hu, W., Qi, Y., Pan, X., Sun, Y., Wang, Z., Liu, C.-Q., Jiang, G., and Fu, P.: High Molecular Diversity of Organic Nitrogen in Urban Snow in North China, Environ. Sci. Technol., 55, 4344-4356, https://dx.doi.org/10.1021/acs.est.0c06851, 2021.

Taguchi, Y.-h. and Oono, Y.: Relational patterns of gene expression via non-metric multidimensional scaling analysis, Bioinformatics, 21, 730-740, 10.1093/bioinformatics/bti067, 2004.

---

## Author Comment (AC2)

**General.**

We would like to appreciate the editor and reviewers for providing the valuable comments and a better perspective on our work to improve the manuscript. In particular, we are very grateful to the editor and reviewers for giving us the opportunity to make revision. We have revised our manuscript by fully taking the reviewers' comments into account. Responses to specific comments raised by the reviewers are described below. All the changes made and appeared in the revised text are shown in red. All detailed answers to comments are displayed in blue.

**Comments of Referee #4 and our responses to them**

Comments:

*The study by Ma et al. deals with investigations of nitrogen-organic compounds in the aqueous phase with different emission scenarios from residential heating and measures. Individual classes of NOC were obtained from LC-qTOF analysis using ESI and complementary supporting information including ions, acids, aerosol liquid water content and meteorological data. Basics concepts from high-resolution mass spectrometry were applied to group NOC according to their elemental composition (CHN, CHON) and polarity of the detection (+/-). By using correlation analysis and literature knowledge of NOC formation, the relative importance of formation pathways of NOC in the aqueous aerosol phase were revealed.*

*The study is generally well-written, despite being lengthy in some sections, and of interest for the readership of Atmospheric Chemistry and Physics. After appropriately addressing few comments below, I would recommend this manuscript for publication.*

Response: We sincerely appreciate your professional and constructive review of our manuscript. Your valuable feedback has greatly improved the clarity and quality of

our work. We have carefully revised the manuscript to address the comments.

Specific comments:

1) *Line 111: 9.95 millions is not a density*

Response: You are correct that the value provided refers to the total population rather than the population density. We have revised the sentence accordingly (Lines 112–113).

Lines 112–113: …The city of HEB, with a population of 9.95 million, is situated in the northeastern region of China.

2) *Line 133: What was the rationale to choose this threshold and how would a different threshold affect the outcome of this study, let's say if 50 or 100 μg/m3 would have been used?*

Response: Thank you for your insightful comment. There are indeed global variations in how $PM_{2.5}$ concentrations are used to define clean and polluted days, reflecting different air quality standards and public health priorities. To provide context, **Table 1** summarizes these thresholds from various countries and organizations.

**Table 1.** Global Standards for Classifying Polluted Days

| Category | Country/ Organization | 24-hour $PM_{2.5}$ Threshold ($\mu g\ m^{-3}$) | Reference |
|---|---|---|---|
| Strict Standards | World Health Organization | >15 | WHO Global Air Quality Guidelines (WHO, 2021) |
| Moderate Standards | United States | >35.4 | EPA Air Quality Standards (EPA) |
| | Japan | >35 | Ministry of the Environment, Japan (Japan, 2009) |
| Lenient Standards | China | >75 | Technical Regulation on Ambient Air Quality Index (on trial) (MEEPRC, 2012) |
| | India | >60 | India CPCB Air Standards (India, 2009) |

To address your suggestion and consider global standards, we adopted an additional threshold of 50 µg m$^{-3}$ to classify clean and polluted days, as it represents a midpoint between moderate and lenient standards. The results using the 50 µg m$^{-3}$ threshold are shown in **Fig. SS1**, corresponding to the results in the main text using the 75 µg m$^{-3}$ threshold.

[Figure]

**Figure SS1**. Box and whisker plots showing variations in the concentration of (**a**) PM$_{2.5}$, (**b**) SO$_2$, and (**c**) nss-Cl$^-$ in all (gray), clean (blue), and haze (red) periods in different cities. Each box encompasses the 25th–75th percentiles. Whiskers are the 5th and 95th percentiles. The green squares and solid lines inside boxes indicate the mean and median value. The consumption of (**d**) raw coal and (**e**) natural gas in 2017 and 2018 in different cities was obtained from the local statistical yearbooks. Average distributions in the signal intensity of species detected in PM$_{2.5}$ collected during different winter periods in different cities in (**f**) ESI+ and (**g**) ESI− modes. Percentage variations in the signal intensity of each subgroup from clean to haze periods in different cities in (**h**) ESI+ and (**i**) ESI− modes.

The results demonstrate that the results from a threshold of 50 µg m$^{-3}$ were comparable to those obtained with the 75 µg m$^{-3}$ threshold. This consistency suggests that the choice of threshold does not substantially affect the overall conclusions of our study.

Since our study focuses on densely populated Chinese cities, we adopted the 75 µg m$^{-3}$ threshold to ensure consistency with widely accepted practices in China (Zhang and Cao, 2015; Xu et al., 2024; Yan et al., 2024). This choice aligns our findings with the national air quality standards and facilitates meaningful comparisons with other studies conducted in similar contexts. We have provided a more detailed explanation of the criteria for selecting clean and polluted days in the revised manuscript (Lines 132–138).

Lines 132–138: …In China, according to the Air Quality Index (MEEPRC, 2012), a pollution day is defined as a day with a 24-hour average PM$_{2.5}$ concentration above 75 µg m$^{-3}$. This standard has also been used in other studies performed in China (Zhang and Cao, 2015; Xu et al., 2024; Yan et al., 2024), showing that the sampling periods were classified as either "clean" or "haze" based on whether the daily average concentration of PM$_{2.5}$ was below or above 75 µg m$^{-3}$.

3) *Line 148: 0.1% formic acid solution has a pH<3, which may potentially lead to hydrolysis or acid-catalyzed reactions. How can you be sure that you observation are not an artifact of you sample workflow?*

Response: Thank you for your insightful comment. Firstly, formic acid is a widely used additive in liquid chromatography (LC) due to its ability to enhance analytical sensitivity (Núñez and Paolo, 2014; Kuehnbaum and Britz-McKibbin, 2013), particularly in electrospray ionization (ESI) mode (Cech and Enke, 2001; Hamilton et al., 2006). It also improves chromatographic performance by facilitating the separation of complex samples and increasing analyte detection sensitivity

(Kuehnbaum and Britz-McKibbin, 2013; Gao et al., 2005).

Secondly, the aerosol particles investigated in this study were inherently acidic (generally pH<3). Furthermore, the LC gradient elution time in this study was only 18 minutes, significantly shorter than both the sampling duration and the aging time of the particles. This short elution time minimizes the likelihood of significant chemical alterations (e.g., hydrolysis or acid-catalyzed reactions) that could lead to artifacts.

Lastly, numerous laboratory (Zhao et al., 2018; Zhang et al., 2016; Zhang et al., 2015; Witkowski and Gierczak, 2014; Reinnig et al., 2008; Müller et al., 2009) and field studies (Zhang et al., 2024; Abudumutailifu et al., 2024; Wang et al., 2021b) have employed formic acid in LC mobile phases without reporting significant artifacts or adverse effects. This extensive use underscores the robustness of this approach.

Thus, we believe that the acidic mobile phase had negligible impact on the overall measurement results.

4) *Line 168: Were only the mass resolution exploited to calculated sum formula or additional information from retention time or fragmentation spectra? What is the mass resolution of the instrument and how can you be sure that it was sufficient for the analysis? What was the mass range of the detected compounds?*

Response: Thank you for your thoughtful questions regarding the methodology for calculating molecular formulas and evaluating the instrument's mass resolution. We are pleased to provide additional clarification.

In this study, compounds with identical molecular formulas were summed without distinguishing isomers based on retention time or fragmentation spectra. The differentiation of isomers and structural identification using these methods are

beyond the scope of this research. This approach aligns with recent studies (Sun et al., 2024; Jiang et al., 2023; Su et al., 2021) that used FT-ICR MS data to identify major liquid-phase reactions of NOCs through "precursor-product pairing theory," where molecular formulas represent the sum of all isomers detected, as isomers cannot be differentiated by FT-ICR MS. However, we acknowledge that incorporating isomer-specific information and fragmentation spectra could provide valuable insights for more detailed studies in future work.

The Xevo G2-XS QToF-MS (Waters) used in this study has a mass resolving power of ≥40 000 at full width at half maximum (FWHM) (m/z 956) (Wang et al., 2021c; Wang et al., 2021d). Data acquisition and processing were conducted using MassLynx v4.2 software, with a mass tolerance of 5 ppm employed to calculate potential molecular formulas for each ion. Instrument calibration was performed before each analysis to ensure measurement accuracy and reliability. These steps collectively ensure the accuracy and reliability of the measurement data.

Additionally, the mass range of detected compounds (m/z 50–700) (Ma et al., 2024; Ungeheuer et al., 2021; Praplan et al., 2014) is well within the instrument's capability, encompassing small organic acids to larger organic aerosol components in the atmosphere.

For a more detailed explanation of the analytical methods, we have included **Sect. S1. UPLC-ESI-QToFMS Analysis** on the page S3. We have also added the following sentence in the revised manuscript (Lines 158-159): "More detailed information about the UPLC-ESI-QToFMS analysis can be found in **Sect. S1**."

**Sect. S1. UPLC-ESI-QToFMS Analysis**

The samples were analyzed using an Acquity UPLC (Waters) coupled to a Xevo

G2-XS QToF-MS (Waters), which is equipped with an electrospray ionization (ESI) source. The instrument has a mass resolving power of $\geq 40\,000$ at full width at half maximum (FWHM) ($m/z$ 956) (Wang et al., 2021c; Wang et al., 2021d), ensuring high precision in the determination of molecular formulas. The data acquisition and processing were carried out using MassLynx v4.2 software, with a mass tolerance of 5 ppm used to calculate potential molecular formulas for each detected ion. Instrument calibration was performed prior to each analysis to ensure the accuracy and reliability of the measurements.

In this study, compounds with identical molecular formulas were summed without distinguishing isomers based on retention time or fragmentation spectra, as the differentiation of isomers is beyond the scope of this research. This methodology is consistent with the findings of recent studies that have employed Fourier transform-ion cyclotron resonance mass spectrometry (FT-ICR MS) for analysis (Sun et al., 2024; Jiang et al., 2023; Su et al., 2021). The observed molecular formulas represent the cumulative sum of all isomeric species, given the inability of this analytical approach to distinguish between them. The mass range of detected compounds ($m/z$ 50–700) (Ma et al., 2024; Ungeheuer et al., 2021; Praplan et al., 2014) was well within the instrument's capabilities, encompassing a range of compounds from small organic acids to larger secondary organic aerosol components.

5) *Line 325: This section 3.2. is repeated in the following one discussing correlations, could be shortened substantially or even removed for brevity.*

Response: Thank you for your constructive comment. We understand your concern regarding the potential overlap between Sections 3.2 and 3.3. We would like to clarify the distinction between these two sections and explain their progressive relationship:

Section 3.2 focuses on identifying and comparing the potential precursors of aerosol NOCs in different cities, highlighting the role of specific precursor classes (e.g., aromatics, heterocyclics, and aliphatics) in NOC formation. This section provides a detailed overview of precursor contributions in the context of various urban environments.

In contrast, Section 3.3 investigates the main factors influencing aerosol NOC formation, specifically through Spearman correlation analysis and non-metric multidimensional scaling (NMDS) analysis. It examines the key drivers of NOC distribution, including combustion source emissions and secondary aqueous-phase reactions.

While there is some overlap in the discussion of anthropogenic activities and their influence on NOC formation, the two sections approach the topic from distinct angles. Section 3.2 introduces the role of precursors, while Section 3.3 delves into the correlations and factors influencing NOC distribution across cities. Both sections are necessary to provide a comprehensive understanding of the NOC formation process.

To improve the flow and connection between these sections, we have added the following clarifying sentences at the beginning of Section 3.3:

Lines 408-413: As discussed in the previous section, the results indicate that AVOCs play a significant role in the formation of NOCs. Furthermore, secondary processes may contribute to NOC formation to varying extents in different cities. This section provides a detailed discussion of the key factors influencing the molecular distribution of NOCs. First, a Spearman correlation analysis was performed to

examine the relationship between various parameters and NOCs (**Fig. 3** and **Figs. S8−S12**).

These additions are intended to help readers better understand the relationship between the two sections.

6) *Line 415: The Spearman correlation coefficients potentially indicate a moving of primary NOC (of negative correlation) during ageing in the VK space, similar to the common approach by Heald et al. (2010) (doi: 10.1029/2010GL042737). Can we learn anything from the presented data using this concept?*

Response: Thank you very much for highlighting the study by Heald et al. (2010), which provides valuable insights for this study. We have cited this paper and added additional descriptions (Lines 386–389) to address this point.

Lines 386-389: …Concurrently, the $O/C_w$ ratio of aerosol NOCs in HEB was observed to be higher than that of coal-derived aerosols (Song et al., 2018). Heald et al. (2010) previously demonstrated that oxidation processes can lead to an increase in the O:C ratio of organic aerosols.

7) *Line 433: Tracers have known sources and sinks, so organic compounds can only be markers (Noziére et al. (2015), doi: 10.1021/cr5003485). Furthermore, Carlton & Turpin (2013) do not write anything about oxalic acid in the referenced study.*

Response: We agree that organic compounds, including oxalic acid, should be referred to as "markers" rather than "tracers," as emphasized by Nozière et al. (2015).

We have replaced the citation to Carlton & Turpin (2013) with Chen et al. (2021) (doi: 10.1021/acs.est.1c01413), which provides a more appropriate discussion of oxalic acid as a marker for aqueous-phase SOA. The revision has been made in the

revised manuscript (Lines 450–452).

Lines 450–452: …Oxalic acid ($C_2H_2O_4$) has been identified as a marker (defined by Nozière et al. (2015)) for aqueous-phase SOA (Xu et al., 2022; Chen et al., 2021).

8) *Line 453/458: SOA refers to particle formation by gas-to-particle conversion. If the reaction occurs between a (low-volatile) particle constituents and e.g. OH, it belongs to the class of aged-POA.*

Response: Thank you for your insightful comment regarding the definition and distinction between SOA and aged-POA. We would like to provide further clarification based on current literature.

**On the Definition of SOA:** There are indeed varying definitions of Secondary Organic Aerosol (SOA) in the literature. As you mentioned, studies such as Kanakidou et al. (2005) and Ziemann and Atkinson (2012) describe SOA as particulate matter formed through the gas-to-particle conversion of low-volatility organic compounds, which are produced by the gas-phase oxidation of volatile organic compounds (VOCs) with atmospheric oxidants. However, current global models typically underpredict SOA magnitude, distribution, and dynamics, suggesting a limited understanding of their sources and formation processes (Ervens et al., 2011). A growing body of modeling and experimental research (Carlton et al., 2020; Ervens et al., 2011; McNeill, 2015; Lim et al., 2010; Gilardoni et al., 2016; Lamkaddam et al., 2021) suggests that aqueous-phase chemical reactions in cloud droplets and wet aerosols may play a crucial role in SOA formation, representing an important yet previously overlooked pathway. Therefore, SOA can be classified into two types based on its formation mechanism: one formed from VOCs via gas-phase oxidation and gas-to-particle conversion (gasSOA), and another formed through aqueous-phase chemical reactions of precursors in the aerosol phase (aqSOA).

**On Aged-POA:** According to the classic definition, Primary Organic Aerosol (POA) refers to organic compounds directly emitted into the atmosphere. Aged-POA refers to POA that has undergone oxidative aging, but it is still considered part of the primary aerosol due to its initial emission from a source. However, the distinction between aged-POA and SOA has become increasingly blurred in recent studies. For instance, Jimenez et al. (2009) noted that oxidized POA (aged-POA) becomes more similar to SOA, particularly in highly oxidizing environments. Additionally, some aerosol models, such as the Volatility Basis Set (VBS) model (Robinson et al., 2007), show that the volatility and oxidation state of POA can evolve over time, and in some cases, aged-POA can be reclassified as part of SOA. Studies have also shown that POA from fossil fuel combustion (Wang et al., 2021a) and biomass burning (Gilardoni et al., 2016) undergoes aqueous-phase processing that contributes to SOA formation, supporting the view that POA can evolve into SOA via aqueous-phase reactions.

In conclusion, since our manuscript emphasizes the significant role of aqueous-phase reactions in NOCs formation, we adopt a broader and more inclusive definition of SOA. We believe that this expanded perspective better captures the complexity of SOA formation in the atmosphere and aligns with the evolving understanding of SOA processes.

9) *Line 516: In the supplement, the temperature induced de_N pathway (formation of nitriles by reduction of amines) is reasoned with the study by Simoneit et al. (2003) on amides and nitriles from biomass burning, so a primary source. They are proposing that e.g. unsaturated fatty acids are reacting with NH3 in the hot BB exhaust. Explicitly, they preclude that those reactions happen in the atmosphere leading to secondary aerosol formation. I understand the authors that the higher temperature in BJ and HZ is responsible for this mechanism, which I think is a false conclusion.*

**Response:** Thank you for your insightful comment. In response to your concern, we have updated the description to incorporate more recent findings, thereby strengthening our conclusions.

Updated Evidence for Dehydration Reactions in Aerosols:

Recent studies, such as Sun et al. (2024), demonstrate that dehydration reactions involving nitrogen-containing organic compounds (NOCs) can occur in aerosol particles and fog water under ambient, non-high-temperature conditions. Similarly, Lian et al. (2020) observed dehydration reactions during photochemical transformations in aquatic environments, supporting the feasibility of these processes in the particle and aqueous phases. These findings indicate that high temperatures (e.g., those in biomass burning exhaust) are not necessary for these reactions to occur.

Revision of Mechanistic Examples in the Supplement:

We recognize that using amide dehydration as an example may not have been the most representative or accurate illustration of these processes. To improve clarity and accuracy, we have replaced this example with references to studies that explicitly demonstrate dehydration reactions in the aqueous or particle phase under ambient conditions.

Support from Our Data:

Furthermore, our data are consistent with findings from Sun et al. (2024), supporting the hypothesis that dehydration reactions can occur in aerosols under ambient conditions. We agree with your assessment that attributing these reactions solely to higher temperatures in BJ and HZ is not accurate. Instead, we have revised our interpretation of the data to highlight alternative factors. To clarify this point, the revision has been made in the revised manuscript (Lines 532–541).

Lines 532–541: …Recent studies have identified that dehydration reactions may occur in aerosols and fog water (Sun et al., 2024), as well as in photochemical

transformations of organic compounds in aqueous phase (Lian et al., 2020). While the exact pathways of dehydration reactions in the particle phase remain uncertain, stronger solar radiation in BJ and HZ than in HEB (**Table S1**) may partly explain the higher signal proportion of CHON+ compounds formed through the cond_hy_N pathway in BJ and HZ. Furthermore, the higher signal proportions of CHN+ compounds formed through the de_N pathway in BJ (6%) and HZ (11%) than in HEB (2%) may also be associated with this solar radiation-induced dehydration mechanism (**Fig. 5d–f** and **Table S12**).

These revisions ensure that our interpretation of the mechanisms is both clearer and more accurate. We sincerely appreciate your feedback, which has been instrumental in improving the clarity and robustness of our manuscript.

10) *Line 539: Could you exclude that the low correlation for HZ is not caused by generally low intensities affected by the larger relative uncertainty of the measurement?*

Response: Thank you for this insightful question. We have carefully considered the possibility that the low correlation for HZ might be influenced by generally low intensities and the larger relative uncertainty of the measurements. The detailed explanation is as follows.

Measurement Uncertainty and Quality Control: The measurement uncertainties in our study were systematically evaluated and documented. Prior to analysis, we conducted rigorous quality control procedures to ensure data reliability. Specifically, the data used in our analysis were obtained by subtracting the mass spectrometry signals of blank samples from those of environmental samples. Additionally, any sample signals lower than five times the ion signal intensity of the blank sample were excluded from further analysis. These measures ensured that the dataset primarily consisted of robust and reliable signals.

Instrument Sensitivity: The instruments and methods employed in our analysis were highly sensitive and precise, capable of detecting target compounds even at lower concentration levels. For HZ, the signal-to-noise ratios for the key measurements consistently remained within acceptable ranges, indicating that measurement uncertainty did not significantly impact the observed correlations.

Comparison Across Cities: To ensure comparability, we applied the same data quality checks and statistical analyses across all three cities. The relatively low correlation observed for HZ is more likely attributable to differences in emission sources, chemical processes, or environmental factors specific to HZ (e.g., milder winter temperatures and the absence of central heating), rather than measurement-related uncertainties.

We hope this clarifies our assessment. Thank you again for raising this important point.

11) *Line 561: You pointed out that HZ has generally mild winters and the emission of NOC precursors is lower due to generally lower heating activities. Therefore, I do not understand the link to pollution control measures.*

Response: Thank you for your thoughtful comments. We understand your concern regarding the relationship between the absence of centralized heating and the lower emissions of NOC precursors in HZ.

In light of your comments, we have revised the relevant section to better clarify that the mild winter conditions in HZ reduce the need for centralized heating, leading to lower precursor emissions. Additionally, stricter pollution control measures, which are likely more stringent in HZ than in BJ (more coal usage in HZ than in BJ, as shown in **Fig. 1d**), also contribute to the overall lower emissions. This combination

of factors results in less precursor availability for the formation of NOCs in the aqueous phase.

In general, due to generally mild winters leading to the absence of heating and the implementation of stricter pollution control measures (more coal usage in HZ than in BJ, as shown in **Fig. 1d**), the precursor emissions in HZ were lower. These emissions were insufficient to support the production of large amounts of NOCs in the aqueous phase (Lines 583–587).

Lines 583–587: …In contrast, due to generally mild winters leading to the absence of heating and the implementation of stricter pollution control measures (more coal usage in HZ than in BJ, as shown in **Fig. 1d**), the precursor emissions in HZ were lower. These emissions were insufficient to support the production of large amounts of NOCs in the aqueous phase.

We appreciate your constructive feedback, which has helped us improve the clarity of the manuscript. We hope the revised explanation addresses your concerns.

12) *Line 616: Similar to line 561, I think the absent central heating policy is not a reasons for lower NOC formation as the illustration suggests but rather a consequence of a missing need for stricter measures to control emission due to a warmer climate zone.*

Response: Thank you for your insightful comment. The revision has been made in the revised manuscript, as follows.

Lines 622–625: …In contrast, due to generally mild winters resulting in the absence of a winter heating policy and the implementation of strict pollution control measures, as mentioned previously, aromatic precursor emissions in HZ were expected to be the lowest.

**Technical comment:**

*13) Line 433: "Xu et al. (2022a)" does not exist in the reference list, but two times "Xu et al. (2022b)".*

Response: Thank you very much for your meticulous review and for pointing this out. Upon careful re-evaluation of the manuscript and reference list, we would like to clarify that "Xu et al. (2022a)" and "Xu et al. (2022b)" refer to two distinct publications, both of which are cited correctly in the text and listed appropriately in the references. We appreciate your attention to detail and have taken this opportunity to ensure that the references are consistently formatted and clearly presented throughout the manuscript to avoid any potential confusion.

To provide further clarification, Xu et al. (2022a) refers to the study titled "Large contribution of fossil-derived components to aqueous secondary organic aerosols in China" (Lines 1053–1057), while Xu et al. (2022b) refers to "Water-Insoluble Components in Rainwater in Suburban Guiyang, Southwestern China: A Potential Contributor to Dissolved Organic Carbon" (Lines 1062–1065). Both references are distinct and appropriately cited in the manuscript.

We also acknowledge that the first authors of these studies share the same surname, "Xu," but have different given names, which may have led to some confusion. We have carefully reviewed the manuscript to ensure that all citations are accurate and clearly distinguishable.

For your reference, the full citations are provided below:

Xu, B., Zhang, G., Gustafsson, Ö., Kawamura, K., Li, J., Andersson, A., Bikkina, S., Kunwar, B., Pokhrel, A., Zhong, G., Zhao, S., Li, J., Huang, C., Cheng, Z., Zhu, S., Peng, P., and Sheng, G.: Large contribution of fossil-derived components to

aqueous secondary organic aerosols in China, Nature Communications, 13, 5115, https://doi.org/10.1038/s41467-022-32863-3, 2022a.

Xu, Y., Dong, X.-N., Xiao, H.-Y., He, C., and Wu, D.-S.: Water-Insoluble Components in Rainwater in Suburban Guiyang, Southwestern China: A Potential Contributor to Dissolved Organic Carbon, J. Geophys. Res.-Atmos., 127, e2022JD037721, https://doi.org/10.1029/2022JD037721, 2022b.

**At last, we deeply appreciate the time and effort you've spent in reviewing our manuscript.**

**Reference:**

Abudumutailifu, M., Shang, X., Wang, L., Zhang, M., Kang, H., Chen, Y., Li, L., Ju, R., Li, B., Ouyang, H., Tang, X., Li, C., Wang, L., Wang, X., George, C., Rudich, Y., Zhang, R., and Chen, J.: Unveiling the Molecular Characteristics, Origins, and Formation Mechanism of Reduced Nitrogen Organic Compounds in the Urban Atmosphere of Shanghai Using a Versatile Aerosol Concentration Enrichment System, Environ. Sci. Technol., 10.1021/acs.est.3c04071, 2024.

Carlton, A. G., Christiansen, A. E., Flesch, M. M., Hennigan, C. J., and Sareen, N.: Multiphase Atmospheric Chemistry in Liquid Water: Impacts and Controllability of Organic Aerosol, Accounts of Chemical Research, 53, 1715-1723, 10.1021/acs.accounts.0c00301, 2020.

Cech, N. B. and Enke, C. G.: Practical implications of some recent studies in electrospray ionization fundamentals, Mass Spectrometry Reviews, 20, 362-387,

https://doi.org/10.1002/mas.10008, 2001.

Chen, Y., Guo, H., Nah, T., Tanner, D. J., Sullivan, A. P., Takeuchi, M., Gao, Z., Vasilakos, P., Russell, A. G., Baumann, K., Huey, L. G., Weber, R. J., and Ng, N. L.: Low-Molecular-Weight Carboxylic Acids in the Southeastern U.S.: Formation, Partitioning, and Implications for Organic Aerosol Aging, Environ. Sci. Technol., 55, 6688-6699, 10.1021/acs.est.1c01413, 2021.

EPA: United States Environmental Protection Agency: AQI Breakpoints: https://aqs.epa.gov/aqsweb/documents/codetables/aqi_breakpoints.html, last access: December 10, 2024,

Ervens, B., Turpin, B. J., and Weber, R. J.: Secondary organic aerosol formation in cloud droplets and aqueous particles (aqSOA): a review of laboratory, field and model studies, Atmos. Chem. Phys., 11, 11069-11102, 10.5194/acp-11-11069-2011, 2011.

Gao, S., Zhang, Z.-P., and Karnes, H. T.: Sensitivity enhancement in liquid chromatography/atmospheric pressure ionization mass spectrometry using derivatization and mobile phase additives, Journal of Chromatography B, 825, 98-110, https://doi.org/10.1016/j.jchromb.2005.04.021, 2005.

Gilardoni, S., Massoli, P., Paglione, M., Giulianelli, L., Carbone, C., Rinaldi, M., Decesari, S., Sandrini, S., Costabile, F., Gobbi, G. P., Pietrogrande, M. C., Visentin, M., Scotto, F., Fuzzi, S., and Facchini, M. C.: Direct observation of aqueous secondary organic aerosol from biomass-burning emissions, P. Natl. Acad. Sci. USA, 113, 10013-10018, doi:10.1073/pnas.1602212113, 2016.

Hamilton, J. F., Lewis, A. C., Reynolds, J. C., Carpenter, L. J., and Lubben, A.: Investigating the composition of organic aerosol resulting from cyclohexene ozonolysis: low molecular weight and heterogeneous reaction products, Atmos. Chem. Phys., 6, 4973-4984, 10.5194/acp-6-4973-2006, 2006.

Heald, C. L., Kroll, J. H., Jimenez, J. L., Docherty, K. S., DeCarlo, P. F., Aiken, A. C., Chen, Q., Martin, S. T., Farmer, D. K., and Artaxo, P.: A simplified description of the evolution of organic aerosol composition in the atmosphere, Geophys. Res. Lett., 37, https://doi.org/10.1029/2010GL042737, 2010.

India, C.: National Ambient Air Quality Standards: http://www.indiaenvironmentportal.org.in/files/Air%20pollution%20note_final.pdf, last access: December 10, 2024, 2009.

Japan, M. o. t. E.: Environmental Quality Standards for the PM2.5, last access: December 10, 2024, 2009.

Jiang, H., Cai, J., Feng, X., Chen, Y., Wang, L., Jiang, B., Liao, Y., Li, J., Zhang, G., Mu, Y., and Chen, J.: Aqueous-Phase Reactions of Anthropogenic Emissions Lead to the High Chemodiversity of Atmospheric Nitrogen-Containing Compounds during the Haze Event, Environ. Sci. Technol., 57, 16500-16511, 10.1021/acs.est.3c06648, 2023.

Jimenez, J. L., Canagaratna, M. R., Donahue, N. M., Prevot, A. S. H., Zhang, Q., Kroll, J. H., DeCarlo, P. F., Allan, J. D., Coe, H., Ng, N. L., Aiken, A. C., Docherty, K. S., Ulbrich, I. M., Grieshop, A. P., Robinson, A. L., Duplissy, J., Smith, J. D., Wilson, K. R., Lanz, V. A., Hueglin, C., Sun, Y. L., Tian, J., Laaksonen, A.,

Raatikainen, T., Rautiainen, J., Vaattovaara, P., Ehn, M., Kulmala, M., Tomlinson, J. M., Collins, D. R., Cubison, M. J., E., Dunlea, J., Huffman, J. A., Onasch, T. B., Alfarra, M. R., Williams, P. I., Bower, K., Kondo, Y., Schneider, J., Drewnick, F., Borrmann, S., Weimer, S., Demerjian, K., Salcedo, D., Cottrell, L., Griffin, R., Takami, A., Miyoshi, T., Hatakeyama, S., Shimono, A., Sun, J. Y., Zhang, Y. M., Dzepina, K., Kimmel, J. R., Sueper, D., Jayne, J. T., Herndon, S. C., Trimborn, A. M., Williams, L. R., Wood, E. C., Middlebrook, A. M., Kolb, C. E., Baltensperger, U., and Worsnop, D. R.: Evolution of Organic Aerosols in the Atmosphere, Science, 326, 1525-1529, doi:10.1126/science.1180353, 2009.

Kanakidou, M., Seinfeld, J. H., Pandis, S. N., Barnes, I., Dentener, F. J., Facchini, M. C., Van Dingenen, R., Ervens, B., Nenes, A., Nielsen, C. J., Swietlicki, E., Putaud, J. P., Balkanski, Y., Fuzzi, S., Horth, J., Moortgat, G. K., Winterhalter, R., Myhre, C. E. L., Tsigaridis, K., Vignati, E., Stephanou, E. G., and Wilson, J.: Organic aerosol and global climate modelling: a review, Atmos. Chem. Phys., 5, 1053-1123, 10.5194/acp-5-1053-2005, 2005.

Kuehnbaum, N. L. and Britz-McKibbin, P.: New Advances in Separation Science for Metabolomics: Resolving Chemical Diversity in a Post-Genomic Era, Chem. Rev., 113, 2437-2468, 10.1021/cr300484s, 2013.

Lamkaddam, H., Dommen, J., Ranjithkumar, A., Gordon, H., Wehrle, G., Krechmer, J., Majluf, F., Salionov, D., Schmale, J., Bjelić, S., Carslaw, K. S., El Haddad, I., and Baltensperger, U.: Large contribution to secondary organic aerosol from isoprene cloud chemistry, Science Advances, 7, eabe2952, doi:10.1126/sciadv.abe2952,

2021.

Lian, L., Yan, S., Zhou, H., and Song, W.: Overview of the Phototransformation of Wastewater Effluents by High-Resolution Mass Spectrometry, Environ. Sci. Technol., 54, 1816-1826, 10.1021/acs.est.9b04669, 2020.

Lim, Y. B., Tan, Y., Perri, M. J., Seitzinger, S. P., and Turpin, B. J.: Aqueous chemistry and its role in secondary organic aerosol (SOA) formation, Atmos. Chem. Phys., 10, 10521-10539, 10.5194/acp-10-10521-2010, 2010.

Ma, Y. J., Xu, Y., Yang, T., Xiao, H. W., and Xiao, H. Y.: Measurement report: Characteristics of nitrogen-containing organics in $PM_{2.5}$ in Ürümqi, northwestern China – differential impacts of combustion of fresh and aged biomass materials, Atmos. Chem. Phys., 24, 4331-4346, 10.5194/acp-24-4331-2024, 2024.

McNeill, V. F.: Aqueous Organic Chemistry in the Atmosphere: Sources and Chemical Processing of Organic Aerosols, Environ. Sci. Technol., 49, 1237-1244, 10.1021/es5043707, 2015.

MEEPRC: Technical Regulation on Ambient Air Quality Index (on trial): HJ 633—2012, Ministry of Ecology and Environment of the People's Republic of China, https://www.mee.gov.cn/ywgz/fgbz/bz/bzwb/jcffbz/201203/t20120302_224166.shtml, (last access: 10 December 2024), 2012.

Müller, L., Reinnig, M.-C., Hayen, H., and Hoffmann, T.: Characterization of oligomeric compounds in secondary organic aerosol using liquid chromatography coupled to electrospray ionization Fourier transform ion cyclotron resonance mass spectrometry, Rapid Commun. Mass Spectrom., 23, 971-979,

https://doi.org/10.1002/rcm.3957, 2009.

Nozière, B., Kalberer, M., Claeys, M., Allan, J., D'Anna, B., Decesari, S., Finessi, E., Glasius, M., Grgić, I., Hamilton, J. F., Hoffmann, T., Iinuma, Y., Jaoui, M., Kahnt, A., Kampf, C. J., Kourtchev, I., Maenhaut, W., Marsden, N., Saarikoski, S., Schnelle-Kreis, J., Surratt, J. D., Szidat, S., Szmigielski, R., and Wisthaler, A.: The Molecular Identification of Organic Compounds in the Atmosphere: State of the Art and Challenges, Chem. Rev., 115, 3919-3983, 10.1021/cr5003485, 2015.

Núñez, O. and Paolo, L.: Applications and uses of formic acid in liquid chromatography-mass spectrometry analysis, in: Advances in Chemistry Research, edited by: Taylor, J. C., Nova Science Publishers, 71-86, 2014.

Praplan, A. P., Hegyi-Gaeggeler, K., Barmet, P., Pfaffenberger, L., Dommen, J., and Baltensperger, U.: Online measurements of water-soluble organic acids in the gas and aerosol phase from the photooxidation of 1,3,5-trimethylbenzene, Atmos. Chem. Phys., 14, 8665-8677, 10.5194/acp-14-8665-2014, 2014.

Reinnig, M.-C., Müller, L., Warnke, J., and Hoffmann, T.: Characterization of selected organic compound classes in secondary organic aerosol from biogenic VOCs by HPLC/MSn, Analytical and Bioanalytical Chemistry, 391, 171-182, 10.1007/s00216-008-1964-5, 2008.

Robinson, A. L., Donahue, N. M., Shrivastava, M. K., Weitkamp, E. A., Sage, A. M., Grieshop, A. P., Lane, T. E., Pierce, J. R., and Pandis, S. N.: Rethinking Organic Aerosols: Semivolatile Emissions and Photochemical Aging, Science, 315, 1259-1262, doi:10.1126/science.1133061, 2007.

Song, J., Li, M., Jiang, B., Wei, S., Fan, X., and Peng, P. a.: Molecular Characterization of Water-Soluble Humic like Substances in Smoke Particles Emitted from Combustion of Biomass Materials and Coal Using Ultrahigh-Resolution Electrospray Ionization Fourier Transform Ion Cyclotron Resonance Mass Spectrometry, Environ. Sci. Technol., 52, 2575-2585, https://doi.org/10.1021/acs.est.7b06126, 2018.

Su, S., Xie, Q., Lang, Y., Cao, D., Xu, Y., Chen, J., Chen, S., Hu, W., Qi, Y., Pan, X., Sun, Y., Wang, Z., Liu, C.-Q., Jiang, G., and Fu, P.: High Molecular Diversity of Organic Nitrogen in Urban Snow in North China, Environ. Sci. Technol., 55, 4344-4356, https://dx.doi.org/10.1021/acs.est.0c06851, 2021.

Sun, W., Hu, X., Fu, Y., Zhang, G., Zhu, Y., Wang, X., Yan, C., Xue, L., Meng, H., Jiang, B., Liao, Y., Wang, X., Peng, P., and Bi, X.: Different formation pathways of nitrogen-containing organic compounds in aerosols and fog water in northern China, Atmos. Chem. Phys., 24, 6987-6999, 10.5194/acp-24-6987-2024, 2024.

Ungeheuer, F., van Pinxteren, D., and Vogel, A. L.: Identification and source attribution of organic compounds in ultrafine particles near Frankfurt International Airport, Atmos. Chem. Phys., 21, 3763-3775, 10.5194/acp-21-3763-2021, 2021.

Wang, J., Ye, J., Zhang, Q., Zhao, J., Wu, Y., Li, J., Liu, D., Li, W., Zhang, Y., Wu, C., Xie, C., Qin, Y., Lei, Y., Huang, X., Guo, J., Liu, P., Fu, P., Li, Y., Lee, H. C., Choi, H., Zhang, J., Liao, H., Chen, M., Sun, Y., Ge, X., Martin, S. T., and Jacob, D. J.: Aqueous production of secondary organic aerosol from fossil-fuel emissions in winter Beijing haze, P. Natl. Acad. Sci. USA, 118, e2022179118,

doi:10.1073/pnas.2022179118, 2021a.

Wang, K., Huang, R.-J., Brueggemand, M., Zhang, Y., Yang, L., Ni, H., Guo, J., Wang, M., Han, J., Bilde, M., Glasius, M., and Hoffmann, T.: Urban organic aerosol composition in eastern China differs from north to south: molecular insight from a liquid chromatography-mass spectrometry (Orbitrap) study, Atmos. Chem. Phys., 21, 9089-9104, https://doi.org/10.5194/acp-21-9089-2021, 2021b.

Wang, Y., Zhao, Y., Li, Z., Li, C., Yan, N., and Xiao, H.: Importance of Hydroxyl Radical Chemistry in Isoprene Suppression of Particle Formation from α-Pinene Ozonolysis, ACS Earth Space Chem., 5, 487-499, https://doi.org/10.1021/acsearthspacechem.0c00294, 2021c.

Wang, Y., Zhao, Y., Wang, Y., Yu, J. Z., Shao, J., Liu, P., Zhu, W., Cheng, Z., Li, Z., Yan, N., and Xiao, H.: Organosulfates in atmospheric aerosols in Shanghai, China: seasonal and interannual variability, origin, and formation mechanisms, Atmos. Chem. Phys., 21, 2959-2980, 10.5194/acp-21-2959-2021, 2021d.

WHO: WHO global air quality guidelines: https://www.who.int/publications/i/item/9789240034228, last access: 10 December 2024, 2021.

Witkowski, B. and Gierczak, T.: Early stage composition of SOA produced by α-pinene/ozone reaction: α-Acyloxyhydroperoxy aldehydes and acidic dimers, Atmos. Environ., 95, 59-70, https://doi.org/10.1016/j.atmosenv.2014.06.018, 2014.

Xu, B., Zhang, G., Gustafsson, Ö., Kawamura, K., Li, J., Andersson, A., Bikkina, S.,

Kunwar, B., Pokhrel, A., Zhong, G., Zhao, S., Li, J., Huang, C., Cheng, Z., Zhu, S., Peng, P., and Sheng, G.: Large contribution of fossil-derived components to aqueous secondary organic aerosols in China, Nature Communications, 13, 5115, 10.1038/s41467-022-32863-3, 2022.

Xu, Y., Liu, T., Ma, Y. J., Sun, Q. B., Xiao, H. W., Xiao, H., Xiao, H. Y., and Liu, C. Q.: Measurement report: Occurrence of aminiums in PM2.5 during winter in China – aminium outbreak during polluted episodes and potential constraints, Atmos. Chem. Phys., 24, 10531-10542, 10.5194/acp-24-10531-2024, 2024.

Yan, F., Su, H., Cheng, Y., Huang, R., Liao, H., Yang, T., Zhu, Y., Zhang, S., Sheng, L., Kou, W., Zeng, X., Xiang, S., Yao, X., Gao, H., and Gao, Y.: Frequent haze events associated with transport and stagnation over the corridor between the North China Plain and Yangtze River Delta, Atmos. Chem. Phys., 24, 2365-2376, 10.5194/acp-24-2365-2024, 2024.

Zhang, M., Cai, D., Lin, J., Liu, Z., Li, M., Wang, Y., and Chen, J.: Molecular characterization of atmospheric organic aerosols in typical megacities in China, npj Climate and Atmospheric Science, 7, 230, 10.1038/s41612-024-00784-1, 2024.

Zhang, X., Dalleska, N. F., Huang, D. D., Bates, K. H., Sorooshian, A., Flagan, R. C., and Seinfeld, J. H.: Time-resolved molecular characterization of organic aerosols by PILS + UPLC/ESI-Q-TOFMS, Atmos. Environ., 130, 180-189, https://doi.org/10.1016/j.atmosenv.2015.08.049, 2016.

Zhang, X., McVay, R. C., Huang, D. D., Dalleska, N. F., Aumont, B., Flagan, R. C., and Seinfeld, J. H.: Formation and evolution of molecular products in α-pinene

secondary organic aerosol, P. Natl. Acad. Sci. USA, 112, 14168-14173, doi:10.1073/pnas.1517742112, 2015.

Zhang, Y.-L. and Cao, F.: Fine particulate matter (PM$_{2.5}$) in China at a city level, Scientific Reports, 5, 14884, 10.1038/srep14884, 2015.

Zhao, R., Kenseth, C. M., Huang, Y., Dalleska, N. F., and Seinfeld, J. H.: Iodometry-Assisted Liquid Chromatography Electrospray Ionization Mass Spectrometry for Analysis of Organic Peroxides: An Application to Atmospheric Secondary Organic Aerosol, Environ. Sci. Technol., 52, 2108-2117, 10.1021/acs.est.7b04863, 2018.

Ziemann, P. J. and Atkinson, R.: Kinetics, products, and mechanisms of secondary organic aerosol formation, Chemical Society Reviews, 41, 6582-6605, 10.1039/c2cs35122f, 2012.

---

## Author Response (AR2)

**General.**

We would like to appreciate the editor for providing the valuable comments on our work. We have revised our manuscript by fully taking the editor's comments into account. Responses to specific comments raised by the editor are described below. All the changes made and appeared in the revised text are shown in red. All detailed answers to comments are displayed in blue.

**Comments of the editor and our responses to them**

Comments:

*I would like to thank the authors for taking the Reviewers' comments seriously and having revised the manuscript accordingly.*

Response: We appreciate your professional review for our article. We have revised the manuscript to address the comments. Our responses to the specific comments and changes made in the manuscript are given below.

Specific comments:

1) *For the comment about the potential influence of using 0.1% formic acid. The authors argued that the particles are inherently acidic (generally pH<3), which needs more support. In Northern China, particles are generally less acidic nowadays. CHON compounds which can go hydrolysis could be potentially affected by the low pH.*

Response: We appreciate the editor's comments regarding pH conditions and the potential hydrolysis of CHON compounds. Below, we address the concerns in detail:

1. Acidity of aerosols in Northern China

According to recent studies (Wang et al., 2021b; Zhang et al., 2023a; Li et al., 2024; Wang et al., 2023; Zhang et al., 2023b), aerosols during haze episodes in Northern China typically were acidic, with pH values ranging from 4 to 5. These findings were similar to our observations in Harbin and Beijing during haze periods. In contrast, in Hangzhou, aerosol acidity was higher (an average pH value of 3) (Li et al., 2025; Nah et al., 2023). These results indicate that aerosols remain acidic or weakly acidic during winter haze periods in our study regions.

2. Potential impact of pH on CHON compound hydrolysis

We recognize that CHON compounds susceptible to hydrolysis might be influenced by low pH conditions. However, the likelihood of such reactions significantly altering our results is minimized for the following reasons:

The LC gradient elution time in this study was only 18 minutes, which was substantially shorter than the sampling duration and aging time of the particles. This short elution time reduced the probability of significant chemical changes to the compounds.

Additionally, numerous laboratory and field studies (Zhao et al., 2018; Zhang et al., 2016; Zhang et al., 2024; Abudumutailifu et al., 2024; Wang et al., 2021a) using formic acid in LC mobile phases have not reported substantial artifacts or adverse effects on analytes under similar conditions.

In response to these concerns, we have provided additional clarification in **Sect. S1. UPLC-ESI-QToFMS Analysis** in the Supporting Information (SI). The added content is as follows (Pages S3-S4):

The addition of formic acid to the mobile phase played a crucial role in optimizing chromatographic separation and enhancing ionization efficiency during ESI– MS analysis (Núñez and Paolo, 2014; Kuehnbaum and Britz-Mckibbin, 2013). This

approach is commonly employed in the analysis of atmospheric organic compounds (Zhang et al., 2024; Abudumutailifu et al., 2024; Wang et al., 2021a). Although the acidic conditions introduced by formic acid may affect certain CHON compounds through hydrolysis or acid-catalyzed reactions, the aerosols analyzed in this study were generally acidic or mildly acidic (average pH values of 3–5). Additionally, the short elution time (18 minutes) minimized the likelihood of significant chemical changes to the compounds.

2) *Line 192, change to "CHN+ compounds"*

Response: The revision has been made in the revised manuscript (Line 192).

3) *Line 264, change "contributed" to "contribute"*

Response: The revision has been made in the revised manuscript (Line 264).

4) *Line 278, use "mean values"*

Response: The revision has been made in the revised manuscript (Line 278).

5) *Line 583-585, change this sentence to "In contrast, due to the absence of heating for generally mild winters and the implementation of stricter pollution control measures"*

Response: We greatly appreciate your suggestions. The revision has been made in the revised manuscript (Lines 583-584).

**At last, we deeply appreciate the time and effort you've spent in reviewing our manuscript.**

**Reference:**

Abudumutailifu, M., Shang, X., Wang, L., Zhang, M., Kang, H., Chen, Y., Li, L., Ju, R., Li, B., Ouyang, H., Tang, X., Li, C., Wang, L., Wang, X., George, C., Rudich, Y., Zhang, R., and Chen, J.: Unveiling the Molecular Characteristics, Origins, and Formation Mechanism of Reduced Nitrogen Organic Compounds in the Urban Atmosphere of Shanghai Using a Versatile Aerosol Concentration Enrichment System, Environ. Sci. Technol., 10.1021/acs.est.3c04071, 2024.

Kuehnbaum, N. L. and Britz-McKibbin, P.: New Advances in Separation Science for Metabolomics: Resolving Chemical Diversity in a Post-Genomic Era, Chem. Rev., 113, 2437-2468, 10.1021/cr300484s, 2013.

Li, G., Su, H., Zheng, G., Zhou, M., Han, W., Zhang, Y., Ma, N., Wang, H., Klimach, T., and Cheng, Y.: Novel Device for in Situ and Real-Time Detection of the Acidity of Ambient Aerosols: Laboratory Characterization and Ambient Measurements, Environ. Sci. Technol., 59, 659-667, 10.1021/acs.est.4c09221, 2025.

Li, W., Qi, Y., Liu, Y., Wu, G., Zhang, Y., Shi, J., Qu, W., Sheng, L., Wang, W., Zhang, D., and Zhou, Y.: Daytime and nighttime aerosol soluble iron formation in clean and slightly polluted moist air in a coastal city in eastern China, Atmos. Chem. Phys., 24, 6495-6508, 10.5194/acp-24-6495-2024, 2024.

Nah, T., Lam, Y. H., Yang, J., and Yang, L.: Long-term trends and sensitivities of PM2.5

pH and aerosol liquid water to chemical composition changes and meteorological parameters in Hong Kong, South China: Insights from 10-year records from three urban sites, Atmos. Environ., 302, 119725, https://doi.org/10.1016/j.atmosenv.2023.119725, 2023.

Núñez, O. and Paolo, L.: Applications and uses of formic acid in liquid chromatography-mass spectrometry analysis, in: Advances in Chemistry Research, edited by: Taylor, J. C., Nova Science Publishers, 71-86, 2014.

Wang, K., Huang, R.-J., Brueggemand, M., Zhang, Y., Yang, L., Ni, H., Guo, J., Wang, M., Han, J., Bilde, M., Glasius, M., and Hoffmann, T.: Urban organic aerosol composition in eastern China differs from north to south: molecular insight from a liquid chromatography-mass spectrometry (Orbitrap) study, Atmos. Chem. Phys., 21, 9089-9104, https://doi.org/10.5194/acp-21-9089-2021, 2021a.

Wang, T., Liu, Y., Zhou, S., Wang, G., Liu, X., Wang, L., Fu, H., Chen, J., and Zhang, L.: Key Factors Determining the Formation of Sulfate Aerosols Through Multiphase Chemistry—A Kinetic Modeling Study Based on Beijing Conditions, J. Geophys. Res.-Atmos., 128, e2022JD038382, https://doi.org/10.1029/2022JD038382, 2023.

Wang, W., Liu, M., Wang, T., Song, Y., Zhou, L., Cao, J., Hu, J., Tang, G., Chen, Z., Li, Z., Xu, Z., Peng, C., Lian, C., Chen, Y., Pan, Y., Zhang, Y., Sun, Y., Li, W., Zhu, T., Tian, H., and Ge, M.: Sulfate formation is dominated by manganese-catalyzed oxidation of SO2 on aerosol surfaces during haze events, Nature Communications, 12, 1993, 10.1038/s41467-021-22091-6, 2021b.

Zhang, M., Cai, D., Lin, J., Liu, Z., Li, M., Wang, Y., and Chen, J.: Molecular characterization of atmospheric organic aerosols in typical megacities in China, npj Climate and Atmospheric Science, 7, 230, 10.1038/s41612-024-00784-1, 2024.

Zhang, X., Dalleska, N. F., Huang, D. D., Bates, K. H., Sorooshian, A., Flagan, R. C., and Seinfeld, J. H.: Time-resolved molecular characterization of organic aerosols by PILS + UPLC/ESI-Q-TOFMS, Atmos. Environ., 130, 180-189, https://doi.org/10.1016/j.atmosenv.2015.08.049, 2016.

Zhang, X., Tong, S., Jia, C., Zhang, W., Wang, Z., Tang, G., Hu, B., Liu, Z., Wang, L., Zhao, P., Pan, Y., and Ge, M.: Elucidating HONO formation mechanism and its essential contribution to OH during haze events, npj Climate and Atmospheric Science, 6, 55, 10.1038/s41612-023-00371-w, 2023a.

Zhang, Y., Chen, Y., Jiang, N., Wang, S., Zhang, R., Lv, Z., Hao, X., and Wei, Y.: Chemical-composition characteristics of PM1 and PM2.5 and effects on pH and light-extinction coefficients under different pollution levels in Zhengzhou, China, Journal of Cleaner Production, 409, 137274, https://doi.org/10.1016/j.jclepro.2023.137274, 2023b.

Zhao, R., Kenseth, C. M., Huang, Y., Dalleska, N. F., and Seinfeld, J. H.: Iodometry-Assisted Liquid Chromatography Electrospray Ionization Mass Spectrometry for Analysis of Organic Peroxides: An Application to Atmospheric Secondary Organic Aerosol, Environ. Sci. Technol., 52, 2108-2117, 10.1021/acs.est.7b04863, 2018.